# Rethinking Scale-Aware Temporal Encoding for Event-based Object Detection

**Lin Zhu**[1], **Tengyu Long**[1], **Xiao Wang**[2], **Lizhi Wang**[3], **Hua Huang**[3,*]

[1] School of Computer Science, Beijing Institute of Technology
[2] School of Computer Science and Technology, Anhui University
[3] School of Artificial Intelligence, Beijing Normal University
linzhu@bit.edu.cn, longtengyu@bit.edu.cn
xiaowang@ahu.edu.cn, wanglizhi@bnu.edu.cn, huahuang@bnu.edu.cn

## Abstract

Event cameras provide asynchronous, low-latency, and high-dynamic-range visual signals, making them ideal for real-time perception tasks such as object detection. However, effectively modeling the temporal dynamics of event streams remains a core challenge. Most existing methods follow frame-based detection paradigms, applying temporal modules only at high-level features, which limits early-stage temporal modeling. Transformer-based approaches introduce global attention to capture long-range dependencies, but often add unnecessary complexity and overlook fine-grained temporal cues. In this paper, we propose a CNN-RNN hybrid framework that rethinks temporal modeling for event-based object detection. Our approach is based on two key insights: (1) introducing recurrent modules at lower spatial scales to preserve detailed temporal information where events are most dense, and (2) utilizing Decoupled Deformable-enhanced Recurrent Layers specifically designed according to the inherent motion characteristics of event cameras to extract multiple spatiotemporal features, and performing independent downsampling at multiple spatiotemporal scales to enable flexible, scale-aware representation learning. These multi-scale features are then fused via a feature pyramid network to produce robust detection outputs. Experiments on Gen1, 1 Mpx and eTram dataset demonstrate that our approach achieves superior accuracy over recent transformer-based models, highlighting the importance of precise temporal feature extraction in early stages. This work offers a new perspective on designing architectures for event-driven vision beyond attention-centric paradigms.

Code: `https://github.com/BIT-Vision/SATE`

## 1 Introduction

Event cameras provide a fundamentally different visual sensing modality by asynchronously recording pixel-level brightness changes with microsecond latency [8]. Their high dynamic range, sparse output, and low power consumption make them well-suited for real-time perception in high-speed or low-light environments [46]. These characteristics render event cameras particularly attractive for performing visual tasks such as object detection [27, 31], tracking [9, 39], and optical flow estimation [44, 6, 42], even in challenging scenarios (e.g., extreme lighting variations and high-speed motion dynamics).

Event data is stored as asynchronous arrays containing the spatial coordinates, polarity, and timestamp of each illumination change. In contrast, conventional frame-based cameras represent visual information through fixed-rate pixel value matrices. This fundamental discrepancy renders conventional

---

[*]Corresponding author.

39th Conference on Neural Information Processing Systems (NeurIPS 2025).

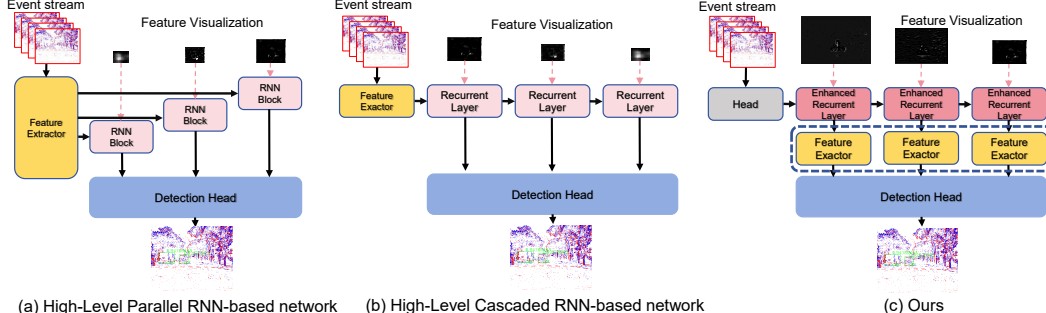

(a) High-Level Parallel RNN-based network     (b) High-Level Cascaded RNN-based network     (c) Ours

Figure 1: Comparison of different temporal modeling strategies in event-based object detection. Specifically, (a) adopts a *High-Level Parallel RNN-based network* (e.g., DMANet [38]), where RNN modules are independently inserted at different feature scales. (b) shows a *High-Level Cascaded RNN-based network* (e.g., RED[31], RVT [11], SAST [29]), in which RNN modules are stacked sequentially across multiple scales to achieve cascaded temporal aggregation. In contrast, (c) illustrates *Ours*, where enhanced recurrent modules are inserted *before significant downsampling*. Multi-scale features are extracted separately via scale-specific branches. Here, RNN Block refers to basic recurrent units such as ConvLSTM [33], while Recurrent Layer combines recurrent modeling with additional feature extraction operations. The proposed Enhanced Recurrent Layer design is detailed in Sec. 3.

image-based neural networks incompatible with event data processing. To address this issue, existing methodologies are primarily categorized into two methodological branches. The first approach involves transforming raw spatiotemporal event streams into dense representations analogous to multi-channel images (such as voxel grid [45], event histogram [28], and time surface [20, 34]). Such transformations facilitate the application of established computer vision techniques originally designed for frame-based data. The second category adopts sparse computational paradigms, typically employing spiking neural networks (SNNs) or graph-based architectures, yet these frequently encounter challenges including hardware incompatibility and suboptimal accuracy. In this work, we utilize dense representations for their advantages in effectiveness.

In addition, how to effectively modeling the temporal dynamics of sparse event streams remains an open challenge, particularly for dense prediction tasks such as object detection. Recent approaches for event-based object detection span a diverse range of architectures. Transformer-based methods [11, 30] leverage global attention to capture long-range temporal dependencies but are often computationally expensive and less effective at modeling localized, high-frequency temporal patterns. Spiking neural networks (SNNs) [4, 41, 47] exploit the asynchronous nature of events but suffer from limited representation capacity and optimization difficulty. More recently, state-space models [50, 40] such as Mamba have shown potential for sequential modeling but remain underexplored in the context of low-level event representation. Notably, many of these methods emphasize temporal modeling at deeper layers (Figure 1), where spatial resolution is low and temporal cues may already be degraded.

In this work, we argue that accurate modeling of early-stage temporal features is essential for effective event-based object detection. Since events are inherently sparse and localized in both space and time, the most informative temporal structures often appear at low-level representations. Motivated by this, we propose to revisit convolutional architectures and augment them with effective recurrent modules at early stages to explicitly preserve fine-grained temporal information. Our design is not aimed at merely reducing model complexity, but at improving temporal fidelity in the feature extraction process.

To this end, we introduce a CNN-RNN hybrid network tailored for event-based object detection. The network consists of three key components. First, we place recurrent blocks at lower spatial scales to enhance temporal modeling at early stages, enabling the network to capture fine-grained temporal patterns that are critical for understanding sparse event dynamics. Second, we propose a divide-and-conquer methodology that enhances current event features through decoupled per-pixel motion estimation and spatiotemporal feature fusion. Third, we design a multi-branch backbone with independent temporal downsampling at three resolution levels, allowing flexible and complementary spatiotemporal feature extraction across varying receptive fields. The multiple spatiotemporal features extracted by the backbone then are hierarchically propagated to a detection head incorporating a

Feature Pyramid Network [24], leveraging multi-scale features for accurate and temporally consistent object predictions. Despite its architectural simplicity, our model achieves state-of-the-art performance on the Gen1 dataset, outperforming recent transformer- and SNN-based baselines.

In summary, the contributions of this work are:

(1) We propose a CNN-RNN hybrid architecture for event-based object detection that rethinks temporal modeling by introducing recurrent modules at lower spatial scales, enabling the capture of fine-grained temporal dynamics from sparse event streams where information is most dense.

(2) We propose a divide-and-conquer methodology that enhances current event features through strategic utilization of temporal information, that decouples the estimation of per-pixel motion from the feature fusion. Meanwhile, we design a scale-specific spatiotemporal encoding strategy that performs independent temporal downsampling across three resolution branches, allowing the network to flexibly extract complementary multi-scale features, which are later fused via a Feature Pyramid Network for robust detection.

(3) Extensive experiments on the Gen1, 1Mpx and eTram benchmark demonstrate that our approach achieves state-of-the-art performance, outperforming recent transformer-based and SNN-based methods, validating the effectiveness of early-stage temporal modeling.

## 2  Related Works

Existing event-based object detection methodologies can be broadly categorized into two major branches: sparsity-aware architectures and dense feedforward networks.

**Sparsity-aware architectures** leverage bio-inspired neural designs, notably graph neural networks (GNNs) and spiking neural networks (SNNs), to directly process asynchronous event streams. GNN-based approaches dynamically construct spatiotemporal graphs through event subsampling strategies [32, 10], identifying temporally and spatially proximate nodes and establishing adaptive edge connections. A key challenge is designing graph topologies that enable efficient information propagation across long spatiotemporal ranges while remaining computationally tractable. SNNs, on the other hand, exploit sparse, time-dependent spike-based computation [4, 1], inherently aligning with the event-driven nature of the data. Like RNNs, SNN neurons maintain temporal states, but emit spikes only when membrane potentials exceed a threshold, introducing non-differentiability into the network. While surrogate gradients [26] can circumvent this issue, they often compromise the sparsity advantage. Despite their theoretical suitability, both GNN- and SNN-based methods face practical limitations in terms of hardware dependency and inferior detection accuracy compared to dense alternatives.

**Dense feedforward architectures** represent the second major branch. Early methods convert event streams into fixed-duration frame representations and apply standard image-based detectors to each temporal slice independently [3, 16, 2, 18, 15]. These methods largely ignore temporal continuity and motion cues, making them less effective in cases involving occlusion, low texture, or slow motion. Later approaches integrate CNNs with RNNs [31, 22, 38], combining the spatial efficiency of CNNs with the sequential modeling capability of RNNs, yielding significant performance gains. With the advent of the Vision Transformer (ViT)[7], transformer-based architectures have gained prominence in event-based detection as well, leveraging self-attention to model long-range dependencies. Specifically, RVT [11] employs a recurrent vision transformer that integrates local–global self-attention with lightweight LSTM [13]-based temporal aggregation for efficient spatiotemporal modeling, while GET [30] introduces grouped token mechanisms and dual self-attention to jointly capture spatial, temporal, and polarity cues for enhanced event representation. In contrast, SSM [50] replaces recurrent operations with continuous-time state-space modeling[12, 35], enabling efficient and frequency-robust temporal aggregation across varying inference rates. While effective, transformers often suffer from computational inefficiency due to the sparsity and noise in event data. Efforts such as SAST [29] attempt to mitigate this by pruning irrelevant tokens and windows via attention-based mechanisms. However, these strategies inevitably compromise the model's global receptive field, limiting their effectiveness in complex scenes.

Our investigation reveals that existing CNN- and Transformer-based approaches combined with recurrent modules primarily emphasize high-level temporal modeling. However, due to the inherent sparsity of event data, reducing spatial resolution often results in the loss of crucial details essential

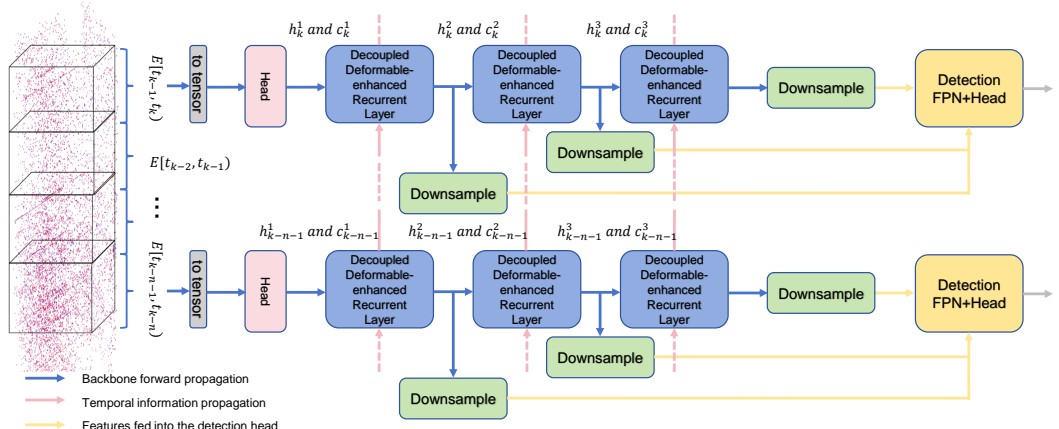

Figure 2: Overview of our designed CNN-RNN network. The event data is first converted into a tensor representation before being fed into the first stage. After passing through a shared shallow stem, the data is processed by our Decoupled Deformable-enhanced Recurrent Layer for spatiotemporal feature extraction. The extracted spatiotemporal features are then further processed in a more flexible manner. Finally, the resulting multi-scale features are fed into the detection head.

for accurate detection. To address this limitation, we strengthen early-stage temporal modeling by integrating event-specific recurrent modules at lower spatial scales, where event activity is denser and temporal cues are richer. Moreover, we introduce a multi-branch downsampling scheme to refine multi-scale spatiotemporal representations, enabling flexible retention of critical information while mitigating information degradation.

## 3 Method

This section presents the proposed CNN-RNN hybrid architecture for event-based object detection. Our design is motivated by two core observations: (1) fine-grained temporal patterns in event data often appear at early stages and are best captured before spatial abstraction; and (2) scale-specific spatiotemporal encoding enables complementary feature extraction across object sizes and motion dynamics. Meanwhile, we adopt a divide-and-conquer approach to decouple feature fusion from motion estimation, further leveraging temporal information, which not only enhances the current frame features, but also leverages the smoothing effect of deformable convolution to effectively suppress the propagation of task-irrelevant information in low-dimensional features. The overall framework is shown in Figure 2.

### 3.1 Event Representation

Event cameras output a stream of asynchronous events represented as tuples $e = (x, y, t, p)$, where $(x, y)$ denotes the spatial location, $t$ is the timestamp, and $p \in \{+1, -1\}$ is the polarity indicating intensity increase or decrease. To make the data compatible with convolutional processing, we discretize the event stream into voxel grids [45] over a short temporal window. The events within a temporal window $\Delta T$ are converted into a voxel grid of size $B \times H \times W$, where $H$ and $W$ denote the height and width of the event frame, respectively, and $B$ represents the number of temporal bins.

$$V(x, y, t) = \sum_i p_i \delta_b(x - x_i) \delta_b(y - y_i) \delta_b(t - t_i^*), \tag{1}$$

where $t_i^* = \frac{B-1}{\Delta T}(t_i - t_1)$ and $\delta_b(a) = max(0, 1 - |a|)$. Our models use a time window $\Delta T = 50ms$ and $B = 5$ temporal bins.

### 3.2 Overall Architecture

The proposed model consists of three major components: Early-stage recurrent encoding for fine-grained temporal modeling, Decoupled Deformable-enhanced Recurrent Layer (DDRL) designed

based on the divide-and-conquer principle and the inherent motion information of event data, Scale-specific spatiotemporal branches with independent downsampling.

As shown in Figure 2, the input voxel slice is processed through a shared shallow stem ($stride = 1$) followed by three successive Decoupled Deformable-enhanced Recurrent Layers with built-in downsampling, yielding spatiotemporal features at three different scales. These features are then fed into three parallel branches, each employing its own temporal encoding mechanism at a specific spatial scale. It is worth noting that each $2\times$ downsampling is accompanied by a doubling of the channel dimensions, except for the final downsampling in each branch, which is intended to reduce the overall model complexity. The outputs of all branches are subsequently passed to the detection head for multi-scale object detection.

### 3.3 Temporal Modeling at Lower Scales

A key challenge in object detection with event cameras lies in whether neural networks can effectively extract meaningful information from both recent events and those generated several seconds earlier. Since event cameras respond to intensity changes caused by object motion, they produce very few events when objects move slowly or remain stationary. This inherent sparsity of event data means that critical information is often embedded in subtle details, which becomes particularly crucial when performing detection across multiple consecutive frames. To preserve this critical temporal information, we place the proposed Decoupled Deformable-enhanced Recurrent Block (DDRB, detailed in Section 3.4) at higher-resolution stages with downsampling factors of 2, 4, and 8. To effectively capture both short-term and long-term dependencies, our model also processes temporal sequences at multiple spatial resolutions. However, unlike previous works [31, 38, 11] emphasizing high-level temporal information propagation, we introduce temporal recurrence *before significant spatial downsampling*, allowing the model to capture dense temporal structures with minimal loss.

In general, the hidden state $H_t^s$ and the cell state $C_t^s$ in recurrent blocks at time $t$ and scale $s$ are updated as:

$$F_t^s = Relu(BN(Conv5 \times 5(F_t^{s-1}))), \tag{2}$$

$$H_t^s, C_t^s = \text{DDRB}(F_t^s, H_{t-1}^s, C_{t-1}^s), \tag{3}$$

where $BN$ means batch normalization [17]. $F_t^{s-1}$ denotes the feature map obtained from the preceding stage, and DDRB represents our proposed Decoupled Deformable-enhanced Recurrent Block, which encapsulates our novel integration of deformable spatial adaptation and temporal recurrence mechanisms.

### 3.4 Decoupled Deformable-enhanced Recurrent Layer

In event-based object detection tasks, event cameras exclusively detect moving objects, leading to suboptimal performance when relying solely on isolated event frames in scenarios involving slow-moving or stationary targets. ConvLSTM [33], a neural network architecture specifically designed for temporal data sequences, addresses this limitation through its capacity for modeling long-range temporal dependencies. This characteristic enables enhanced detection of objects with low-motion or static states within dynamic scenes. Furthermore, the inherent integration of convolutional operations in ConvLSTM facilitates effective processing of spatial features within visual data. Particularly in event-driven object detection, where distinct spatial regions may contain heterogeneous motion patterns, ConvLSTM assists the model in capturing these spatially distributed temporal characteristics, thereby substantially improving detection capabilities. In our proposed Deformable Enhanced Recurrent Layer, we also employ ConvLSTM for spatiotemporal modeling. However, diverging from conventional implementations, we identify that conventional ConvLSTM structures may underutilize extracted spatiotemporal features. To address this, we propose an innovative fusion module based on modulated deformable convolution [48], specifically designed for the characteristics of event data, by adopting a divide-and-conquer strategy. This architectural enhancement enables comprehensive exploitation of spatiotemporal information generated by ConvLSTM, particularly optimizing feature representation through adaptive receptive field adjustment and motion pattern alignment in complex scenarios. Meanwhile, since the recurrent layers are placed at low-dimensional stages, the low-dimensional features may contain task-irrelevant information in addition to the key information we need. The smoothing effect of the deformable convolution can effectively suppress such redundant information.

Specifically, the Decoupled Deformable-enhanced Recurrent Layer (DDRL) comprises a 2× downsampling module and a Decoupled Deformable-enhanced Recurrent Block (DDRB) (as shown in Figure 3). The downsampling module integrates a convolutional layer with kernel size 5 and stride 2, followed by batch normalization (BN) and ReLU activation, to perform preliminary spatial reduction. The DDRB includes a recurrent module (ConvLSTM), and a decoupled modulated deformable convolution specifically designed based on the characteristics of event data. ConvLSTM is employed to capture spatiotemporal information from consecutive event frames, while the decoupled modulated deformable convolution is used to fuse current frame features with historical spatiotemporal representations.

Given the high temporal resolution of event cameras, event data inherently contain fine-grained motion cues. Inspired by this, our design decouples per-pixel motion estimation from feature fusion. We leverage the intrinsic motion information in event data to learn the offsets $\Delta x_k$ and modulated scalars $\Delta m_k$ that 1) adapt the sampling grid of the modulated deformable convolution to pixel-wise motion patterns in the scene, and 2) dynamically weight each sampling positions based on their actual contribution to feature representation. The estimated offsets $\Delta x_k$ and modulation masks $\Delta m_k$ are applied to the features obtained from the preliminary fusion of the current frame and previous spatiotemporal representations, where the fusion is performed by concatenating along the channel dimension followed by a 3×3 convolution for dimensionality reduction. The deformable convolution subsequently adaptively samples features from the compressed representation using the learned $\Delta x_k$ to spatially align with motion patterns, while $\Delta m_k$ dynamically recalibrates sampling weights according to motion relevance. This hierarchical fusion mechanism effectively preserves fine-grained motion cues while eliminating feature redundancy, ensuring optimal utilization of event-driven spatiotemporal correlations.

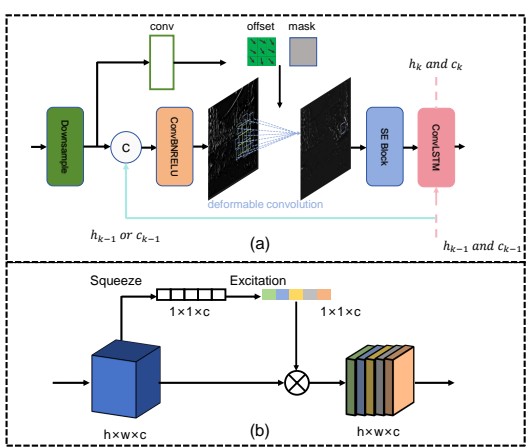

Figure 3: (a) Architecture of the Decoupled Deformable-enhanced Recurrent Layer. Approaches for feature fusion typically employ direct combination through element-wise summation or channel-wise concatenation followed by convolutional integration, or alternatively leverage attention mechanisms for adaptive feature aggregation. Here, we leverage the divide-and-conquer principle to introduce modulated deformable convolution into feature fusion, further exploiting temporal information. (b) Architecture of the SE Block.

Given a convolution kernel with $K$ sampling locations, let $w_k$ and $x_k$ denote the weight and predefined offset of the $k-th$ location, respectively. In this work, we adopt $3 \times 3$ convolution kernel, i.e., $K = 9$ and $x_k \in \{(-1, -1), (-1, 0), \cdots, (1, 1)\}$. At time $t$ and scale $s$, the Decoupled Deformable-enhanced Recurrent Block (DDRB) can be represented as:

$$F_t^{s'} = Relu(BN(Conv3 \times 3(Concat(F_t^s, T_{t-1}^s)))), \tag{4}$$

$$F_t^{s''}(x) = \sum_{k=1}^{K} w_k \cdot F_t^{s'}(x + x_k + \Delta x_k) \cdot \Delta m_k, \tag{5}$$

$$H_t^s, C_t^s = ConvLSTM(SE(F_t^{s''}), H_{t-1}^s, C_{t-1}^s), \tag{6}$$

where $\Delta x_k$ and $\Delta m_k$ represent the learnable offset and modulation scalar for the $k-th$ location, respectively, both obtained from the current event feature $F_t^s$ through two separate $3 \times 3$ convolution layers. $T_{t-1}^s$ denotes the spatiotemporal information from the previous time step $t-1$ ($H_{t-1}^s$ or $C_{t-1}^s$). In the experimental section, we investigate the performance differences when using the hidden state $H_{t-1}^s$, which represents the features of the previous time step, versus the memory cell $C_{t-1}^s$, which stores long-term spatiotemporal information. In addition, to enable the model to flexibly learn diverse deformation patterns across different spatial regions and enhance its representation capability for complex geometric transformations, we adopt grouped deformable convolutions. Specifically, the input channels are divided into multiple groups (set to 8), with each group independently learning distinct deformation patterns. Furthermore, considering that the offset features generated by different groups may contribute unequally to the final task, we incorporate a Squeeze-and-Excitation (SE)

Block [14] after the grouped deformable convolution. The SE Block[14] leverages global average pooling and fully connected layers to learn channel-wise attention weights, thereby enhancing informative channels and suppressing redundant ones, leading to optimized feature representations.

## 3.5 Scale-Specific Spatiotemporal Encoding

Multi-scale feature representation has been extensively employed in object detection frameworks to address scale variation challenges, particularly in detecting objects with diverse sizes. For instance, lower-scale branches preserve higher spatial resolution with shallower network depth, making them particularly suitable for detecting small or rapidly moving objects. As feature resolution progressively decreases through the network hierarchy, the expanded receptive fields enable context-aware detection of larger objects. This multi-scale temporal encoding structure allows the model to learn both fine-grained and coarse-grained motion features. In the present work, we process three low-dimensional multi-scale spatiotemporal features derived from preceding layers using three independent processing branches. Each branch implements scale-specific downsampling operations, followed by feature fusion through a Feature Pyramid Network [24]. This hierarchical architecture enables adaptive resolution adjustment across different network levels, effectively preserving critical information with enhanced flexibility while mitigating information degradation during downsampling.

## 4 Experiment

**Dataset.** The Gen1 automotive dataset [5] consists of 39 hours of event camera recordings with a resolution of $304 \times 240$. It includes $228k$ annotated bounding boxes for vehicles and $28k$ for pedestrians, with available annotation frequencies of 1, 2, or 4 Hz. Following the evaluation protocol of previous works [31, 22, 11], we discard bounding boxes with side lengths smaller than 10 pixels and diagonal lengths shorter than 30 pixels. Similarly, the 1 Mpx dataset [31] focuses on driving scenarios but provides several months of higher-resolution ($1280 \times 720$) daytime and nighttime recordings. It contains approximately 15 hours of event data, annotated at 30 or 60 Hz, with around 25 million bounding box labels distributed across three categories: vehicles, pedestrians, and two-wheelers. We adhere to the same evaluation protocol, removing bounding boxes with side lengths smaller than 20 pixels and diagonal lengths shorter than 60 pixels, and downsample the input resolution to $640 \times 360$. Unlike the Gen1 [5] and 1 Mpx [31] datasets, the eTram dataset [37] is a traffic monitoring dataset collected from a roadside perspective, thus exhibiting higher sparsity. eTram contains approximately 10 hours of data with a resolution of $1280 \times 720$, including around 2 million annotated bounding boxes across 8 categories, with annotations provided at 30 Hz. The preprocessing procedure of the eTram dataset is similar to that of the 1 Mpx dataset. For all datasets, mean Average Precision (mAP) [23] is considered as the primary metric.

**Implementation Details.** During training, we adopt the ADAM optimizer [19] along with a OneCycle [36] learning rate scheduler, which linearly decays from its peak value. Following the strategy in RVT [11], we employ a mixed batch training technique, where standard Backpropagation Through Time (BPTT) is applied to half of the batch samples, while Truncated BPTT (TBPTT) is applied to the other half. Data augmentation includes random horizontal flipping, zoom-in, and zoom-out operations. The event representation is constructed as a 5-channel voxel grid [45] based on a 50 ms time window. For the detection head, we utilize a Feature Pyramid Network (FPN) [24] for multi-scale feature fusion, along with the detection head from YOLOv6 [21], which incorporates distribution focal loss, classification loss, and regression loss.

To compare against prior works on the Gen1 dataset, we train our models with a batch size of 6, sequence length of 21, learning rate of $2e-4$ for $400k$ iterations. The training takes approximately 4 days on a single RTX 3090 GPU. On the 1 Mpx dataset, we train with a batch size of 8, sequence length of 5, and learning rate of $3e-4$ for $800k$ iterations on a single RTX 3090 GPU. On the eTram dataset, the model is trained for $400k$ iterations with a batch size of 8, a sequence length of 5, and an initial learning rate of $3e-4$.

**Benchmark Comparisons.** We compare the proposed neural network architecture with previous works on the Gen1 [5] and 1 Mpx [31] datasets, with the results summarized in Table 1. From Table 1, it can be concluded that the use of recurrent layers generally leads to better performance. S5-ViT-B [50] achieves competitive results by replacing recurrent layers with state-space models (SSM) [12, 35]. ERGO-12 [49] adaptively encodes spatiotemporal information from events, achieving

Table 1: Comparison with state-of-the-art methods on Gen1 and 1 Mpx datasets.

| Method | Params | Backbone | Gen1 mAP | Time (ms) | 1Mpx mAP | Time (ms) |
|---|---|---|---|---|---|---|
| Asynet [26] | 11.4 | Sparse CNN | 14.5 | - | - | - |
| AEGNN [32] | 20.0 | GNN | 16.3 | - | - | - |
| Spiking DenseNet [4] | 8.2 | SNN | 18.9 | - | - | - |
| Inception + SSD [16] | $> 60^*$ | CNN | 30.1 | 19.4 | 34.0 | 45.2 |
| RRC-Events [3] | $> 100^*$ | CNN | 30.7 | 21.5 | 34.3 | 46.4 |
| MatrixLSTM [2] | 61.5 | CNN + RNN | 31.0 | - | - | - |
| YOLOv3 Events [18] | $> 60^*$ | CNN | 31.2 | 22.3 | 31.6 | 49.4 |
| RED [31] | 24.1 | CNN + RNN | 40.0 | 16.7 | 43.0 | 39.3 |
| ASTMNet [22] | $> 100^*$ | CNN + RNN | 46.7 | 35.6 | 48.3 | 72.3 |
| ERGO-12 [49] | 59.6 | Transformer | 50.4 | 69.9 | 46.0 | 100.0 |
| RVT-B [11] | 18.5 | Transformer + RNN | 47.2 | 10.2 | 47.4 | 11.9 |
| Swin-T v2 [25] | 21.1 | Transformer + RNN | 45.5 | 26.6 | 45.5 | 34.8 |
| Nested-T [43] | 22.2 | Transformer + RNN | 46.3 | 20.6 | 46.0 | 33.5 |
| GET-T [30] | 21.9 | Transformer + RNN | 47.9 | 16.8 | 48.4 | 18.2 |
| SAST-CB [29] | 18.9 | Transformer + RNN | 48.2 | 22.7 | 48.7 | 23.6 |
| S5-ViT-B [50] | 18.2 | Transformer + SSM | 47.7 | **8.16** | 47.8 | **9.57** |
| **Ours** | 26.4 | CNN + RNN | **52.7** | 8.80 | **49.1** | 13.3 |

Table 2: Comparison with state-of-the-art methods on the traffic monitoring dataset eTram.

| Method | Backbone | mAP | Time (ms) |
|---|---|---|---|
| RVT-B [11] | Transformer+RNN | 29.5 | **10.88** |
| SAST-CB [29] | Transformer+RNN | 30.0 | 23.07 |
| S5-ViT-B [50] | Transformer+SSM | 29.3 | 14.84 |
| **Ours** | CNN+RNN | **33.0** | 13.05 |

Table 3: Performance of different temporal information reuse methods on the Gen1 test set.

| Fuse-Method | Feature-used | mAP | $AP_{50}$ | Para. (M) |
|---|---|---|---|---|
| Base | - | 50.7 | 80.6 | 22.4 |
| Directly fusion | hidden | 51.7 | 80.8 | 26.6 |
|  | cell | 52.0 | 81.5 |  |
| DDConv | hidden | 52.2 | 81.5 | 26.4 |
|  | cell | 52.7 | 81.5 |  |
| DDConv + SE | hidden | 52.3 | 81.7 | 26.4 |
|  | cell | 52.7 | 81.7 |  |

state-of-the-art performance without employing recurrent layers. It is worth noting that, unlike our CNN-RNN based approach, MatrixLSTM [2] applies LSTM units directly at the input level, while RED [31] and ASTMNet [22] utilize recurrent layers only in deeper network stages.

Our model achieves state-of-the-art performance with an mAP of 52.7 on the Gen1 dataset, and an mAP of 49.1 on the 1 Mpx dataset. Compared to other CNN-RNN methods in the table, our model achieves significantly higher performance while maintaining comparable or lower parameter counts. On the 1 Mpx dataset, it outperforms the second-best ASTMNet [22] by 0.8 mAP, and on the Gen1 dataset, it surpasses it by 6.0 mAP. Notably, our model is trained from scratch without requiring any pre-trained weights.

To further validate the generalizability of our model across datasets, we compared it on the sparser eTram dataset [37] against several Transformer- and SSM-based methods (RVT[11], SAST [29], S5-ViT [50]) that have demonstrated strong performance on the Gen1 [5] and 1 Mpx [31] datasets. As shown in Table 2, our method attains the highest detection accuracy on the eTram dataset [37] while maintaining a high inference speed, achieving an mAP that is 3.0 higher than that of the second-best SAST-CB [29].

**Ablation Study. (1) Effectiveness of Decoupled Deformable-enhanced Recurrent Layer.** Table 3 systematically investigates the effectiveness of our proposed Decoupled Deformable-enhanced Recurrent Block (DDRB). The baseline model ('Base') employs conventional ConvLSTM layers [33] within our custom backbone architecture without temporal enhancement mechanisms, achieving 50.7 mAP on the Gen1 dataset. This baseline performance inherently validates the fundamental efficacy of our backbone design. Furthermore, we attempt to enhance the base model by inserting a

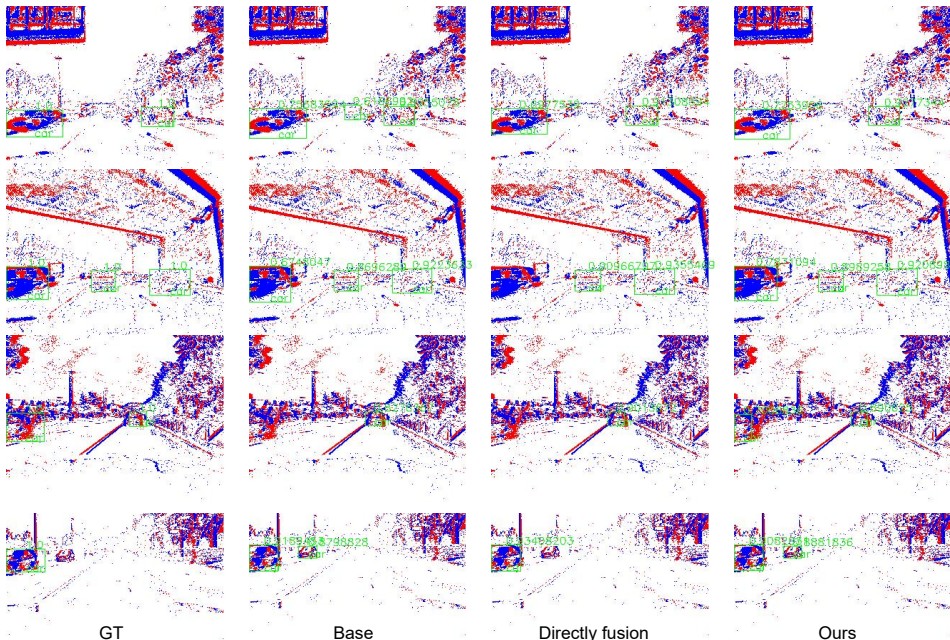

| GT | Base | Directly fusion | Ours |

Figure 4: Visualizations of the Decoupled Deformable-enhanced Recurrent Layer (DDRL) ablation study on the Gen1 dataset.

$5 \times 5$ convolution before each ConvLSTM to fuse the current event features with previous temporal information (called Directly Fusion in Table 3). The results show that using the hidden state leads to a 1.0 mAP improvement, while using the cell state yields a 1.3 mAP increase, indicating that further exploiting prior temporal information can indeed enhance detection accuracy.

Inspired by the divide-and-conquer principle, one of our principal innovations replaces these conventional convolutions with event-based decoupled deformable convolutions (called DDConv in Table 3). This modification achieves superior performance ($+0.5$ mAP with hidden states, $+0.7$ mAP with cell states) while reducing parameter count, establishing an optimal balance between model efficiency and detection accuracy. In addition, we validate the effectiveness of integrating a lightweight SE block [14], which provides further performance improvement without increasing model complexity.

**(2) Effectiveness of the Proposed Backbone.** To further investigate the effectiveness of the proposed backbone, we conducted experiments focusing on its key characteristics (Temporal Modeling at Lower Scales and Scale-Specific Spatiotemporal Encoding). All backbone variants in this study utilized 5-channel event voxel [45] as input and shared a YOLOv6 [21] detection head configuration. We maintained consistent feature downsampling ratios ($8\times$, $16\times$, $32\times$) and corresponding channel dimensions (128, 256, 512) at the detection head input across all configurations. Except for the RVT [11] backbone, all implementations employed cell state for temporal feature enhancement. The experimental results are systematically presented in Table 4.

Our initial investigation explored positioning the Decoupled Deformable-enhanced Recurrent Block (DDRB) in higher-dimensional spaces (as illustrated in Figure 6 (a)). Due to the increased number of channels at higher dimensions, the model parameters surged to 53.8M (an increase of 27.4M). However, this led to a 3.2 mAP drop, indicating that our low-dimensional spatiotemporal modeling strategy fundamentally aligns with event data characteristics. Additionally, we designed a single-branch feature extraction backbone based on our current backbone (with the placement of the recurrent layers unchanged, as shown in Figure 6 (b)). Although this slightly reduced the parameter count, it also led to a 1.4 drop in mAP. Furthermore, we compared our backbone with the RVT [11] backbone one of the representative classical Transformer-RNN methods. Without significantly increasing model complexity, our designed backbone achieved improvements of 4.7 in mAP and 4.2 in $AP_{50}$.

**Visualizations.** Figures 4 and 5 present partial visualization results from the two ablation studies. The comparative analysis demonstrates that our methodology, which strategically leverages temporal information through a divide-and-conquer principle capitalizing on event data characteristics, enhances model robustness for object detection across varying motion velocities. Furthermore,

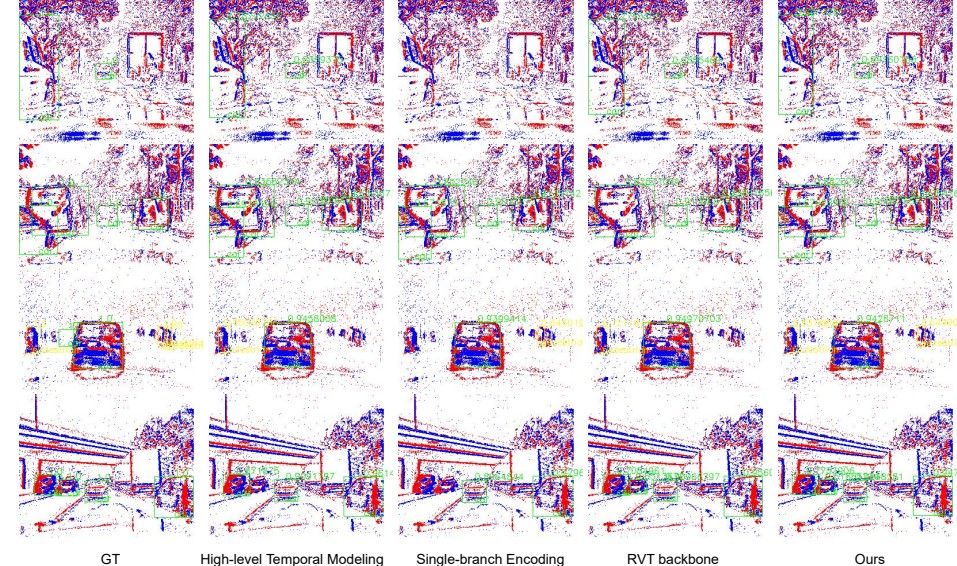

GT  High-level Temporal Modeling  Single-branch Encoding  RVT backbone  Ours

Figure 5: Visualization of the backbone ablation study on the Gen1 dataset.

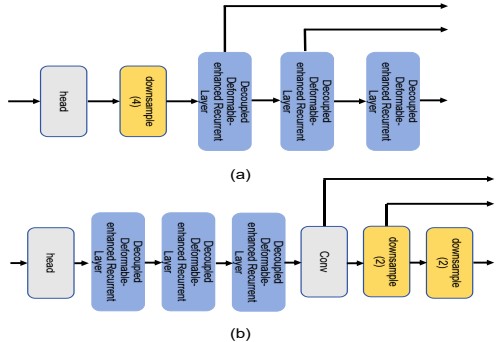

(a)

(b)

Figure 6: Backbone variants used in the ablation studies. (a) High-level Temporal Modeling. (b) Single-branch Encoding. The numbers after "downsample" indicate the downsampling ratio.

Table 4: Performance of different backbone types on the Gen1 test set.

| Backbone Types | mAP | $AP_{50}$ | Para. (M) |
|---|---|---|---|
| High-level Temporal Modeling | 49.5 | 79.5 | 53.8 |
| Single-branch Encoding | 51.3 | 80.9 | 23.5 |
| RVT [11] backbone | 48.0 | 77.5 | 23.2 |
| **Ours** | **52.7** | **81.7** | 26.4 |

the incorporation of fine-grained temporal propagation mechanisms enables better performance in challenging scenarios involving partial occlusions or small object detection.

## 5 Conclusion

This paper revisits the architectural design of event-based object detectors by emphasizing precise temporal modeling over increased complexity. We demonstrate that placing recurrent modules at lower spatial scales enables effective capture of dense temporal patterns present in raw event streams. To further enhance motion alignment and feature quality, we introduce the Decoupled Deformable-enhanced Recurrent Layer (DDRL), which decouples motion estimation from feature fusion and leverages deformable convolution to adaptively align motion while suppressing task-irrelevant noise in low-dimensional features. Combined with scale-specific downsampling and feature fusion, our CNN–RNN architecture achieves competitive or superior performance to transformer-based models without relying on global attention mechanisms. These results highlight that enhancing temporal modeling at the feature extraction stage is key to advancing event-based vision and point toward designing temporally-aware yet structurally simple backbones for sparse, asynchronous data.

## Acknowledgment

This work is partially supported by grants from the National Natural Science Foundation of China under contract No. 62302041.

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

# A Supplementary Experiments

This section primarily provides an additional analysis of the ablation experiments related to the backbone, followed by an investigation of the placement of the Decoupled Deformable-enhanced Recurrent Layer (DDRL) through experimental evaluation.

## A.1 Supplementary Ablation Experiment

During ablation studies on the backbone architecture, we devised two specialized variants, *High-level Temporal Modeling* and *Single-branch Encoding*, explicitly tailored to investigate its critical characteristics: Temporal Modeling at Lower Scales and Scale-Specific Spatiotemporal Encoding. We also compared our backbone with the backbone of the classic Transformer-RNN method RVT [11]. However, we realized that high-dimensional temporal modeling allows for the parallel placement of the recurrent module (as shown in Figure 7 (a)). Therefore, under the same configuration, we conducted an additional experiment, High-level Temporal Modeling (Parallel), where Decoupled Deformable-enhanced Recurrent Block (obtained by removing the downsampling module from DDRL, as shown in Figure 7 (b)) is parallelly placed at high dimensions.

The experimental results are shown in Table 5. Whether the recurrent module is cascaded or parallelly placed at high dimensions, both configurations lead to an increase in parameters accompanied by a performance degradation. This may be because deep networks, through multiple downsampling and filtering operations, may overly smooth sparse event signals, weakening the recurrent module's ability to respond to critical temporal changes. Additionally, the effect of parallel placement of the recurrent module is less effective than cascading, possibly because in the parallel structure, each scale's recurrent module independently processes temporal information, lacking cross-scale temporal context transmission. The high-level semantic features are unable to leverage the fine-grained temporal variations in the low-level high-resolution features, resulting in fragmented temporal modeling. In our approach, recurrent layers are continuously positioned at lower-dimensional stages, aiming to preserve fine-grained temporal information and cross-scale sequential context. Simultaneously, a multi-branch feature extraction mechanism is introduced to flexibly retain salient information while mitigating the loss incurred during the downsampling process.

Table 5: Performance of different backbone types on the Gen1 [5] test set.

| Backbone-Types | mAP | $AP_{50}$ | Para. (M) |
|---|---|---|---|
| High-level Temporal Modeling | 49.5 | 79.5 | 53.8 |
| High-level Temporal Modeling (Parallel) | 47.5 | 77.4 | 53.8 |
| Single-branch Encoding | 51.3 | 80.9 | 23.5 |
| RVT [11] backbone | 48.0 | 77.5 | 23.2 |
| **Ours** | **52.7** | **81.7** | 26.4 |

## A.2 Analysis of the placement of the Decoupled Deformable-enhanced Recurrent Layer (DDRL)

To further explore the optimal placement of the Decoupled Deformable-enhanced Recurrent Layer, we attempted to place it consecutively at different scales (as shown in Figure 8). In the experiment, we still use 5-channel event voxels [45] as input, with FPN [24] performing multi-scale feature fusion and the YOLOv6 [21] detection head conducting detection. Additionally, we ensure that the feature downsampling factors for each backbone input to the detection head are 8, 16, and 32, corresponding to channel numbers of 128, 256, and 512, respectively. The models were trained on the Gen1 [5] dataset for 5 epochs with a batch size of 6 and learning rate of 2e-4.

The experimental results are shown in Table 6. *Low*, *Mid*, and *High* represent the placement of the Decoupled Deformable-enhanced Recurrent Layer at three different positions, from low-dimensional to high-dimensional. The numbers in parentheses indicate the downsampling factors corresponding to the features input to the three ConvLSTM [33]. For example, when placed at the lowest dimension, the downsampling factors for the features input to ConvLSTM are 2, 4, and 8, respectively. It can be observed that placing the Decoupled Deformable-enhanced Recurrent Layer at low dimensions

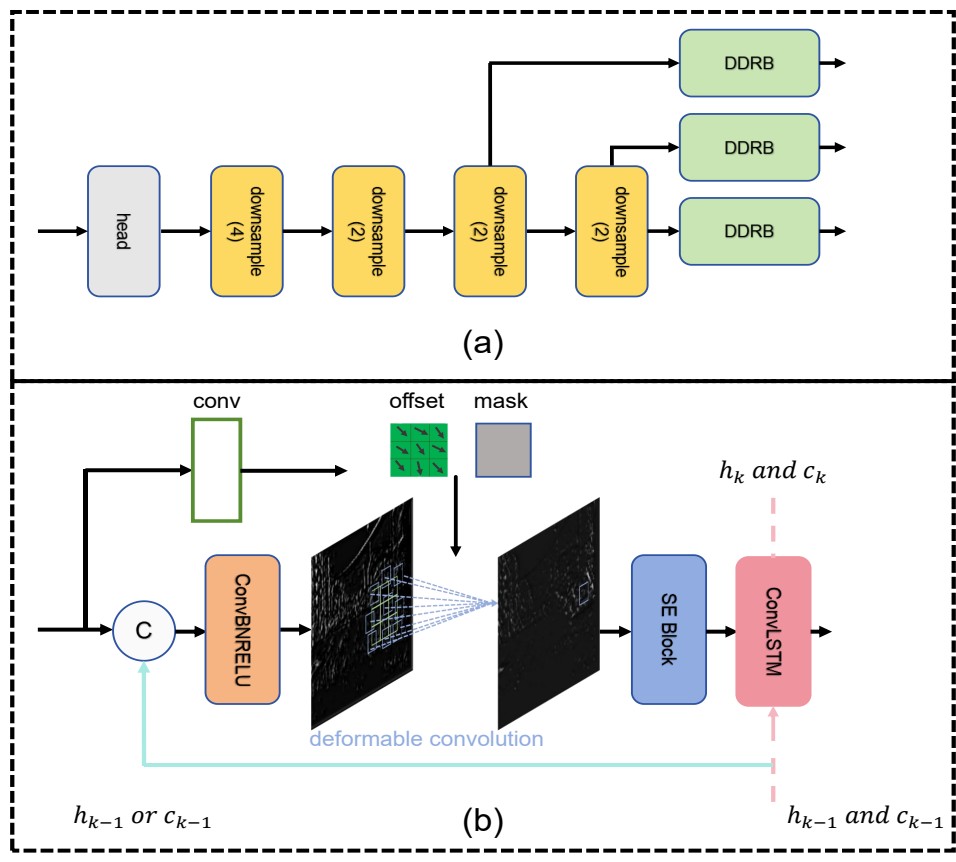

Figure 7: (a) High-level Temporal Modeling (Parallel). The number after 'downsample' indicates the downsampling ratio. (b) The architecture of Decoupled Deformable-enhanced Recurrent Block (DDRB).

yields better performance. This is likely because low-dimensional features typically preserve higher spatial resolution and finer-grained temporal information. Event camera data is inherently sparse and consists of asynchronous event streams. Shallow recurrent modules can directly model short-term motion patterns (such as edge movement and local brightness changes) on low-level abstract features, avoiding temporal blur in deep features caused by multiple downsampling operations.

Table 6: Performance difference of DDRL at different positions on the Gen1 [5] test set.

| Placement | mAP | $AP_{50}$ | Para. (M) |
|---|---|---|---|
| High (8, 16, 32) | 49.5 | 79.5 | 53.8 |
| Mid (4, 8, 16) | 51.0 | 80.1 | 22.9 |
| **Low (2, 4, 8)** | **52.0** | **81.5** | 23.5 |

# B  Visualization

**Visualization of temporal features.** To more intuitively validate the effectiveness of our Decoupled Deformable-enhanced Recurrent Layer for temporal feature enhancement, we visualize the features output by the first and second recurrent layers (as shown in the Figure 9 and 10) and compare them with our base model (which uses ConvLSTM in the recurrent layers without enhancement). We did not choose to visualize the features from the third recurrent layer because, after a certain degree of downsampling, the features became too abstract. It can be observed that, since both models contain recurrent layers, the model is able to retrieve information from past events, even when objects

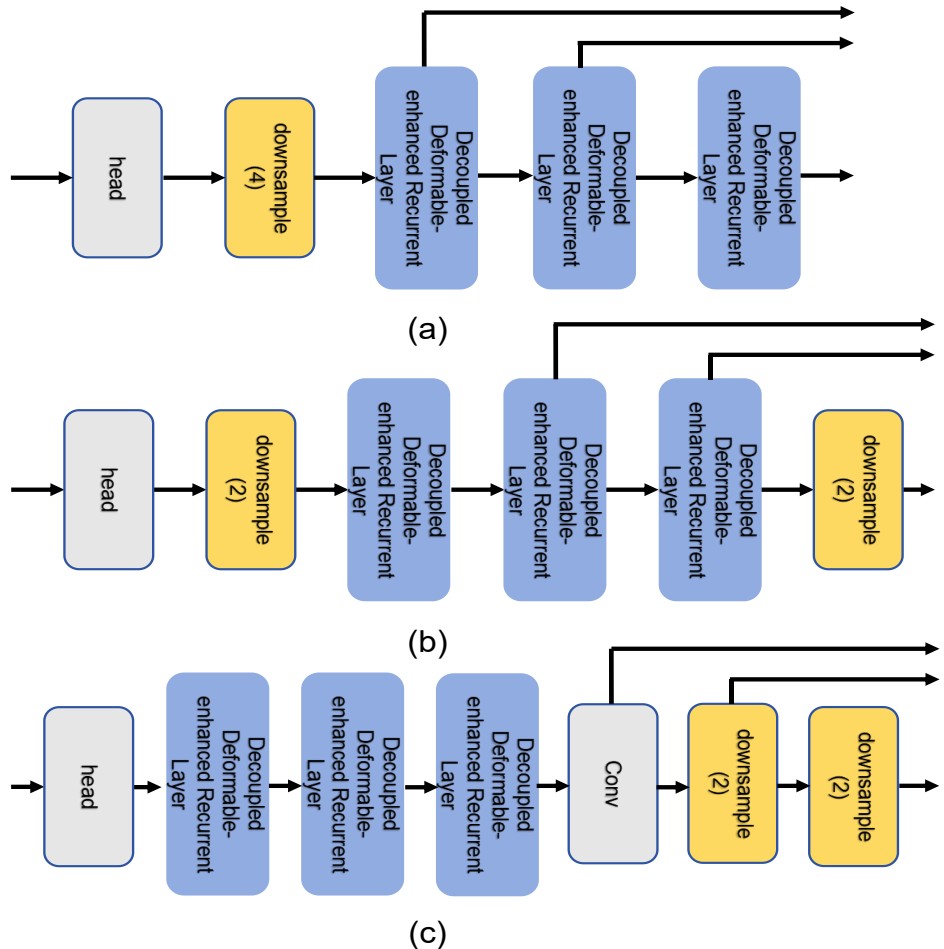

Figure 8: (a), (b), and (c) represent the placement of the Decoupled Deformable-enhanced Recurrent Layer from high-dimensional to low-dimensional, corresponding to *High*, *Mid*, and *Low*, respectively. The number after 'downsample' indicates the downsampling ratio. The 'Conv' used in (c) is specifically employed for channel dimension reduction.

gradually disappear in some scenarios. However, from the lower-dimensional features output by the first recurrent layer, it is evident that after enhancing the temporal features with our method, the details of the features are richer and the noise is significantly reduced. This may be the result of combining the motion information from events with deformable convolutions. From the higher-dimensional features output by the second recurrent layer, it is apparent that, compared to the base model, the enhanced model is able to focus more on the regions where objects are present.

As mentioned in the previous work RED [31], one of the important reasons for placing recurrent layers at high scales is to prevent recurrent layers from dynamically modeling low-level features that are unnecessary for the given task. However, the decoupled deformable-enhanced module we added before the recurrent layers can greatly reduce these unnecessary features through the smoothing effect of deformable convolution.

**Visualization of deformable offsets.** To further validate that the proposed Decoupled Deformable-enhanced Recurrent Layer (DDRL) effectively learns motion-aware spatial alignment, we visualize the learned deformable offsets ($\Delta x_k$) from the first DDRL using heat maps. The visualization is performed on the eTram dataset [37], a traffic monitoring benchmark where the event camera remains almost stationary during recording. This static setup allows us to more intuitively observe the motion patterns of moving vehicles and pedestrians without interference from ego-motion.

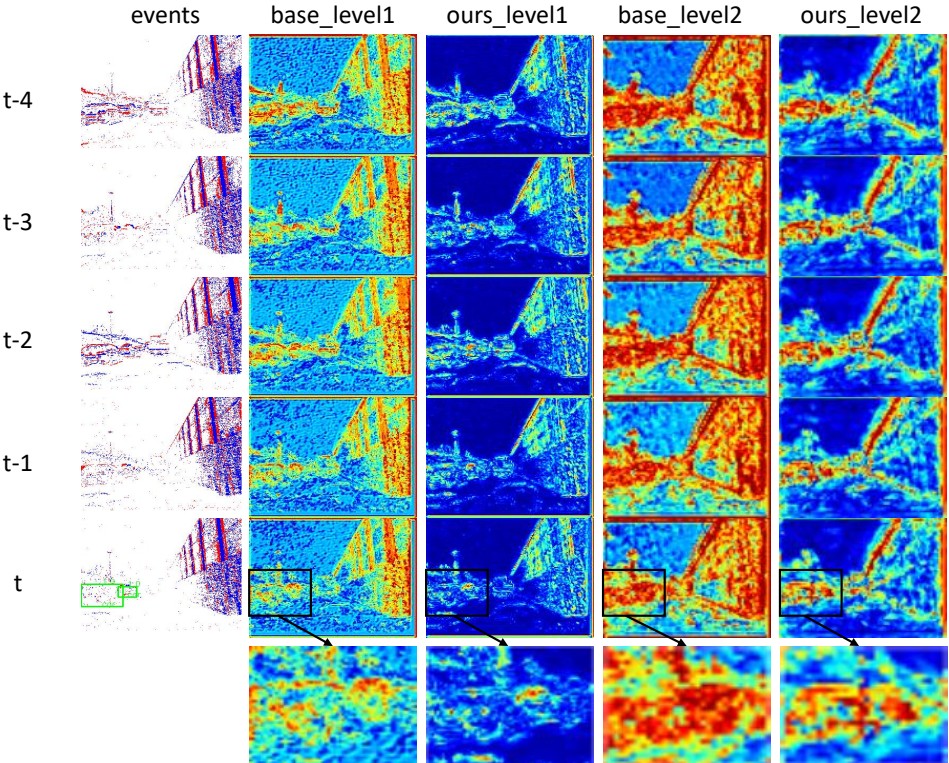

Figure 9: Visualization heatmap of the features output by the first (level1) and second (level2) recurrent layers on Gen1 [5]. 'base' is the variant of our model where temporal enhancement is not applied in the recurrent layers.

As illustrated in Figure 11, the learned offsets exhibit consistent and coherent motion directions aligned with object trajectories, confirming that the deformable module adaptively follows scene dynamics. Specifically, regions corresponding to moving vehicles show large, structured offsets oriented along the motion paths, while static background areas exhibit minimal displacement. This behavior demonstrates that the DDRL effectively decouples per-pixel motion estimation from feature fusion, adaptively aligning spatiotemporal features across consecutive event frames. These results empirically support our design motivation: the deformable convolution in DDRL captures fine-grained motion cues and spatially aligns event features according to actual object motion, thereby enhancing temporal consistency and detection accuracy in dynamic scenes.

## C   Limitations

Although the model we designed is capable of effectively leveraging the sparsity of events as well as the temporal and motion information contained within the events, resulting in promising performance, it also introduces a certain degree of computational complexity. Whether methods such as sparse convolutions can be employed to reduce the computational load without compromising performance is a question we are currently considering.

Moreover, we employ a relatively simple event representation that, although containing some temporal information, does not fully exploit the potential of event-based data. In future work, we aim to explore an event representation incorporating richer temporal information, thereby reducing the dependency of event based object detection on the parameter count and complexity of recurrent layers.

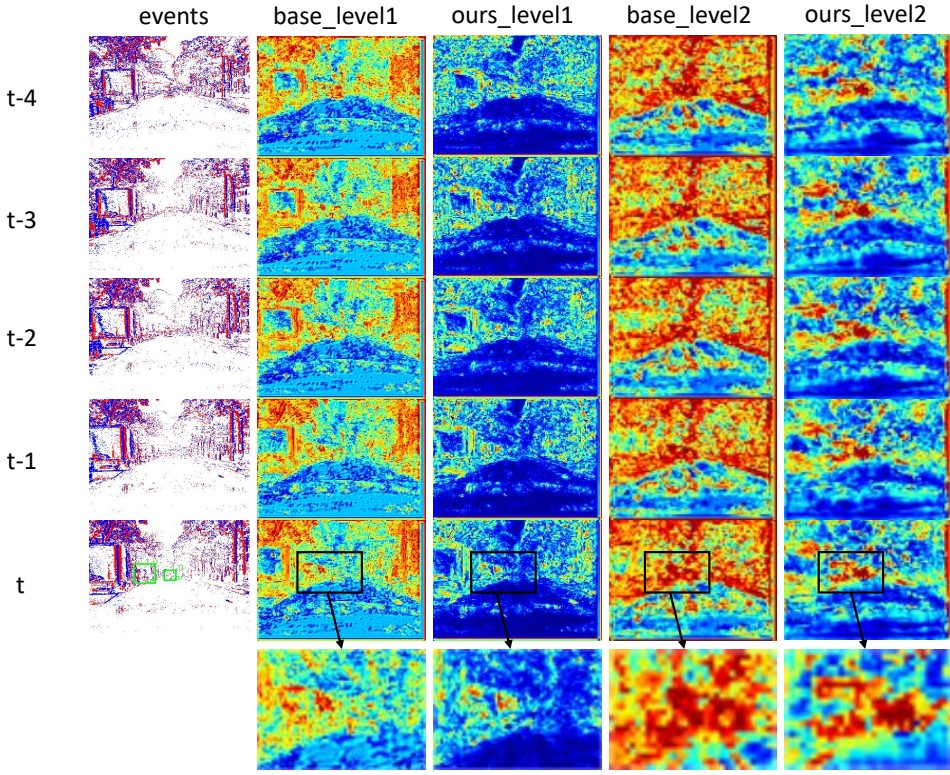

Figure 10: Visualization heatmap of the features output by the first (level1) and second (level2) recurrent layers on Gen1 [5]. 'base' is the variant of our model where temporal enhancement is not applied in the recurrent layers.

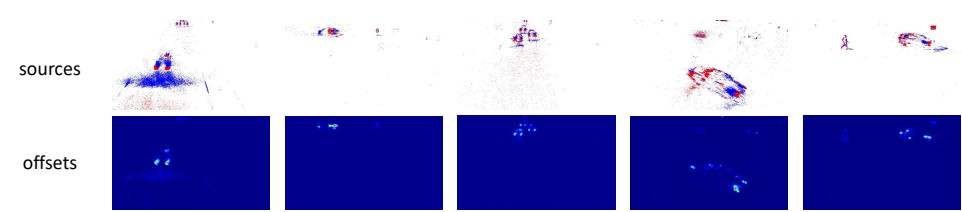

Figure 11: Visualization of the learned offsets from the first DDRL layer along with the corresponding source voxel visualization on eTram [37].

