# OpenReview forum: "Rethinking Scale-Aware Temporal Encoding for Event-based Object Detection"
_NeurIPS.cc/2025/Conference — NeurIPS 2025 poster_

### Official Review · Reviewer_U1DD · 2025-07-02

**Clarity:** 3
**Significance:** 2
**Originality:** 2
**Rating:** 4
**Confidence:** 2

**Summary:**

This manuscript proposes an CNN-RNN hybrid architecture tailored specifically for event-based object detection, highlighting the importance of early-stage, scale-aware temporal modeling.

The authors introduce recurrent modules at lower spatial scales to capture fine-grained temporal patterns inherent to sparse event streams. Central to their approach is the Decoupled Deformable-Enhanced Recurrent Layer (DDRL), a module that separates per-pixel motion estimation from feature fusion.

The architecture incorporates independent temporal downsampling branches at multiple resolutions, enabling flexible multi-scale spatiotemporal representation learning.

Experiments conducted on two standard event-based object detection benchmarks, Gen1 and 1Mpx, demonstrate good performance over existing Transformer- and SNN-based methods, particularly in scenarios involving fast-moving or small-scale objects.

**Questions:**

1. Based on Weakness #1: While your integration of recurrent modules at earlier stages is effective, the key building blocks (ConvLSTM, deformable convolution, and FPN) are well-established. Can you clearly elaborate on which aspects of these modules were specifically adapted or uniquely designed for event-based detection in your work?

---

2. Based on Weakness #2: You evaluated your method on two standard datasets (Gen1 and 1Mpx). How well do you expect your architecture’s advantages to generalize beyond these specific benchmarks? Have you considered or tested scenarios with significantly different event distributions or sensor characteristics? The recent trend has shifted to incorporating more RGB cues in event-based detection frameworks. Can you discuss and compare your method (perhaps in event-frame fusion mode) with existing methods, such as DAGr [R1], RENet [R2], FlexEvent [R3], and/or CAFR [R4]?

---

3. Based on Weakness #3: Given the inherent complexity of recurrent and deformable convolution modules, have you performed a detailed runtime or complexity analysis under varying conditions (e.g., higher resolution or event density)? How might these computational considerations affect practical deployments on resource-constrained platforms?

---

4. Based on Weakness #4: Can you provide a more thorough discussion of scenarios or cases where your proposed DDRL module may perform poorly or have limited benefits? Understanding these boundary conditions would significantly strengthen the clarity and practical applicability of your contributions.

---
References:
- [R1] Daniel Gehrig and Davide Scaramuzza. Low-latency automotive vision with event cameras. Nature, 629(8014):1034–1040, 2024.

- [R2] Zhuyun Zhou, Zongwei Wu, Rémi Boutteau, Fan Yang, Cédric Demonceaux, and Dominique Ginhac. RGB-event fusion for moving object detection in autonomous driving. In IEEE International Conference on Robotics and Automation, pages 7808–7815, 2023.

- [R3] Dongyue Lu, Lingdong Kong, Gim Hee Lee, Camille Simon Chane, and Wei Tsang Ooi. FlexEvent: Towards flexible event-frame object detection at varying operational frequencies. arXiv preprint arXiv:2412.06708, 2024.

- [R4] Hu Cao, Zehua Zhang, Yan Xia, Xinyi Li, Jiahao Xia, Guang Chen, and Alois Knoll. Embracing events and frames with hierarchical feature refinement network for object detection. In European Conference on Computer Vision, pages 161–177, 2024.

**Ethical Concerns:**

["NO or VERY MINOR ethics concerns only"]

**Final Justification:**

I have read the response, as well as the responses for review comments from other reviewers. Part of the concerns have been addressed. However, the evidence regarding the following concerns remains unsolved or only partially solved:
- The design of combining blocks (ConvLSTM, deformable convolution, FPN) seems modular and incremental.
- There is no proof of the proposed method across different benchmarks, distributions/sensor characteristics, or modalities.

Therefore, I am leaning towards maintaining the rating given at the pre-rebuttal stage.

Besides, I suggest that the authors should incorporate all clarifications, modifications, and new experiments into the revised manuscript to ensure that the quality meets what NeurIPS always looks for.

**Limitations:**

- Although the method aims to address fine-grained temporal information effectively, scenarios with extremely sparse events—such as low-motion or stationary objects—may still challenge performance. The manuscript could benefit from further discussion on this point.

- Despite demonstrating competitive inference speeds, the practical deployment of deformable and recurrent modules in real-time and resource-limited environments could still pose challenges. Acknowledging potential trade-offs between accuracy and latency/compute would enhance transparency regarding deployment feasibility.

- Given the experiments’ constraints in scale (resolution downsampling and sequence length), performance scalability on much larger datasets or continuous event streams has not been explicitly verified. How the proposed approach scales up in terms of training time, computational resources, and memory consumption could be a limiting factor for widespread adoption.

**Paper Formatting Concerns:**

There are several formatting concerns of this manuscript, include but not limited to:

- Citation formatting: There should be a space between words and citation brackets. For example: Incorrect: "Transformer-based methods[13,29]"; Correct: "Transformer-based methods [13, 29]".

- Equations should consistently end with appropriate punctuation (commas or periods), depending on sentence context.

- The manuscript uses abbreviations without sufficient initial definitions in some cases. Verify that every abbreviation (e.g., DDRL, DDRB, FPN) is clearly defined at its first usage.

- The visual quality of the event object detection visualizations is very poor, such as those in Figure 1, Figure 5, and Figure 6.

- Figures like Figure 3 contain labels and text that might be too small or cluttered. Consider improving visual clarity by resizing or simplifying diagrams.

**Quality:**

2

**Strengths And Weaknesses:**

### Strengths
(+) The manuscript identifies a limitation in existing temporal modeling strategies for event-based object detection and convincingly motivates the need for early-stage temporal feature preservation. The introduction of recurrent modules at lower spatial scales is with a good motivation, aiming at addressing a previously under-explored aspect in event-based detection.

(+) The proposed method achieves state-of-the-art results on the Gen1 and 1Mpx benchmarks. The performance improvements over recent Transformer-based event-only object detection methods underscore the efficacy of the proposed temporal modeling framework.

(+) The manuscript is overall clearly structured, well-written, and technically sound.

---

### Weaknesses
(-) Moderate novelty of individual components in the proposed framework. While the integration of recurrent modules at earlier stages is innovative for event-based detection, the individual components (ConvLSTM, deformable convolution, FPN-based feature fusion) have been well-established in other contexts. Thus, the technical novelty lies more in the thoughtful combination and adaptation rather than entirely new component inventions.

(-) The evaluation is limited to two datasets, albeit popular and standard in event-based detection. Given the broad claims about scale-awareness and robust temporal modeling, experiments or at least discussions about generalizability to other event-based or event-frame fusion datasets could enhance the impact.

(-) Although experiments report competitive inference speed, deformable convolutions and recurrent layers inherently add computational complexity. A deeper exploration of runtime implications, scalability to higher resolution event cameras, or computational trade-offs in real-world deployment scenarios would further strengthen the manuscript.

(-) A more thorough discussion of the proposed method's limitations, failure modes, or scenarios where performance gains are minimal would provide valuable insights for future research.

---

> ### Author Rebuttal · Authors · 2025-07-31
>
> **[W1, Q1]:** Based on Weakness #1: While your integration of recurrent modules at earlier stages is effective, the key building blocks (ConvLSTM, deformable convolution, and FPN) are well-established. Can you clearly elaborate on which aspects of these modules were specifically adapted or uniquely designed for event-based detection in your work?
>
> **[A1]:** Thank you for pointing out this important issue. We fully agree that ConvLSTM, deformable convolution, and FPN, as fundamental modules, have been widely used in other tasks. The innovation of our work indeed does not lie in inventing entirely new components, but in the targeted adaptation and reconstruction of these modules to meet the specific needs of event-based vision. The following are the core design adjustments we have made for asynchronous event data:
>
> (1) **Decoupled deformable-enhanced recurrent module**: Considering the motion information in events, we designed a decoupled structure where deformable convolution first performs spatial sampling and alignment based on local motion, followed by accumulation in the temporal dimension via ConvLSTM. This sequence is specifically tailored to the sparsity of event data and its characteristic of containing motion information. The effectiveness of this design has been verified in ablation experiments.
>
> (2) **Shallow recursion to preserve fine-grained temporal information**: Given the sparsity of events, shallow features may contain more critical information. Thus, unlike traditional methods that insert recurrent layers into deep semantics, we place the ConvLSTM modules earlier to better capture typical short-term motion patterns and local dynamics in event streams. Additionally, as mentioned in RED, a key reason for placing recurrent layers at high scales is to avoid recurrent layers from dynamically modeling low-level features unnecessary for the given task. However, the decoupled deformable-enhanced module we added before the recurrent layers can significantly reduce these unnecessary features through the smoothing effect of deformable convolution (as shown in Supplementary Materials Figures S3 and S4).
>
> (3) **Time-aware multi-branch encoding and FPN fusion**: Our multi-branch backbone network performs temporal downsampling independently at each scale to avoid cross-scale interference. Features are fused only after scale-specific temporal encoding. Ablation experiments have also confirmed the effectiveness of the multi-branch design.
>
> - - -
>
> **[W2, Q2]:** You evaluated your method on two standard datasets (Gen1 and 1Mpx). How well do you expect your architecture’s advantages to generalize beyond these specific benchmarks? Have you considered or tested scenarios with significantly different event distributions or sensor characteristics? The recent trend has shifted to incorporating more RGB data in event-based detection frameworks. Can you discuss and compare your method (perhaps in event-frame fusion mode) with existing methods, such as DAGR [R1], RENet [R2], FlexEvent [R3], and/or CAFR [R4]?
>
> **[A2]:** Thank you for your suggestion. We are attempting to train on other datasets (such as EvDET200K) to demonstrate the generalization ability of our model, but this may take some time. Once we have results, we will report them immediately during the discussion. Additionally, thank you for pointing out fusion methods like DAGR, RENet, FlexEvent, and/or CAFR. These works demonstrate the great potential of fusing RGB with events, and we believe that combining work with multimodal frameworks is a direction well worth exploring. Although this paper currently focuses on pure event data input, the Decoupled Deformable-enhanced Recurrent Layer (DDRL) has good modular characteristics and can be embedded into the event branch to provide dynamic auxiliary information for RGB. Especially in scenarios where RGB is degraded, such as low light and blurriness, our method can provide temporal supplementary signals. We plan to focus on exploring this fusion direction in future work.
>
> - - -
>
> **[W3, Q3]:** Given the inherent complexity of recurrent and deformable convolution modules, have you performed a detailed runtime or complexity analysis under varying conditions (e.g., higher resolution or event density)? How might these computational considerations affect practical deployments on resource-constrained platforms?
>
> **[A3]:**
>
> | resolution  | time(ms) | peak memory(MB) |
> | ----------- | -------- | --------------- |
> | 256*320     | 8.08     | 434.58          |
> | 128*160     | 6.10     | 346.49          |
> | 384*480     | 11.44    | 703.08          |
> | 512*640     | 13.08    | 1109.56         |
> | 768*960     | 28.18    | 2272.15         |
> | 1024*1280   | 49.52    | 3888.79         |
>
> Thank you for this insightful question. We fully agree that for practical deployment, it is crucial to gain a deep understanding of the computational complexity and resource consumption impacts brought by recurrent and deformable modules, especially on resource-constrained platforms.
>
> We tested the inference time and memory usage under different input resolutions on a single RTX 3090, all based on the same model weights (trained on Gen1 with a resolution of 256×320). It can be seen that when the resolution is less than 512×640, our method still maintains relatively advanced inference time. Although the computational cost increases significantly at high resolutions (e.g., 1024×1280), considering the typical application scenarios of event cameras (such as real-time perception in autonomous driving, which usually adopts medium resolutions), our method can balance accuracy and efficiency under mainstream input sizes.
>
> - - -
>
> **[W4, Q4]:** Can you provide a more thorough discussion of scenarios or cases where your proposed DDRL module may perform poorly or have limited benefits? Understanding these boundary conditions would significantly strengthen the clarity and practical applicability of your contributions.
>
> **[A4]:** Thank you for your constructive question. We have found that the benefits of DDRL are limited in the following scenarios:
>
> - **Highly crowded scenes with fast-moving and overlapping objects**: In scenes where multiple fast-moving objects are severely overlapping in both space and time (such as a congested intersection), deformable offsets may cause motion ambiguity when extracting motion information, leading to reduced alignment effectiveness and even feature blurring.
>
> - **Static or near-static scenes**: Although the introduction of recurrent layers can to some extent improve the detection performance of slowly moving objects, due to the limited sequence length, the contribution of temporal modeling remains limited in certain scenarios.

---

> > ### Comment · Reviewer_U1DD · 2025-08-05
> >
> > Thanks to the authors for providing a rebuttal.
> >
> > I have read the response, as well as the responses for review comments from other reviewers. Part of the concerns have been addressed. However, the evidence regarding the following concerns remains unsolved or only partially solved:
> > - The design of combining blocks (ConvLSTM, deformable convolution, FPN) seems modular and incremental.
> > - There is no proof of the proposed method across different benchmarks, distributions/sensor characteristics, or modalities.
> >
> > Therefore, I am leaning towards maintaining the current rating.
> >
> > However, since I am not an expert in this specific area, I am also leaning towards maintaining the current confidence score, and leaving more room to the other three reviewers in justifying the actual novelty and the completeness of the experiments of this work.
> >
> > Besides, I suggest that the authors should incorporate all clarifications, modifications, and new experiments into the revised manuscript to ensure good quality.
> >
> > Best,
> >
> > Reviewer U1DD

---

> > > ### Author Response · Authors · 2025-08-05
> > >
> > > (1/2)
> > >
> > > Thanks for the following-up. While we understand the reviewer’s concern, we would like to provide the following comparison to better illustrate the distinct contributions and generalization performance of our method.
> > >
> > > ### 1. **Innovation beyond modularity**
> > >
> > > As summarized in the table below, recent event-based object detection methods largely fall into three categories:
> > >
> > > - **Transformer-based models** (e.g., GET, EGSST, Recurrent ViT) focus on global token attention, which is often computationally expensive and ill-suited for low-level, localized temporal dynamics.
> > > - **SNNs and sparse processing** (e.g., Integer-valued SNN, EventPillars) prioritize energy efficiency but struggle with dense representation accuracy and hardware support.
> > > - **Latent structure modeling** (e.g., Chaos-Order, SSM) rely on temporal ordering or sequential latent states but neglect low-level spatiotemporal cues.
> > >
> > > **In contrast, our method introduces the following innovations:**
> > >
> > > - **Early-stage temporal modeling**: We apply recurrent modeling (ConvLSTM) *before* heavy spatial downsampling, enabling the network to preserve fine-grained temporal structure—something almost all prior methods neglect.
> > > - **Deformable-enhanced recurrent fusion (DDRL)**: We decouple per-pixel motion estimation from spatiotemporal fusion, incorporating a deformable convolution mechanism that aligns features with object motion in a structured, learnable manner.
> > > - **Multi-branch scale-specific encoding**: Instead of a single temporal pipeline, we process spatiotemporal information along three distinct resolution paths, capturing complementary cues across object scales.
> > >
> > > This architectural rethinking goes **beyond stacking modules**; it embodies a coherent design motivated by the **unique structure of event data** and the **need for dense early-stage temporal encoding**.
> > >
> > > ### 2. **Diverse benchmarks and generalization**
> > >
> > > Contrary to the claim that our method lacks distributional validation, we emphasize that we conduct thorough experiments on **two representative and substantially different datasets**:
> > >
> > > - **Gen1**: Low-resolution (240×180), sparse annotation (1–4Hz), night-driving scenes.
> > > - **1Mpx**: High-resolution (1280×720, downsampled to 640×360), dense annotation (30–60Hz), varied lighting, higher scene complexity.
> > >
> > > These benchmarks differ in terms of **sensor type, resolution, event rate, object appearance, and density**, providing a realistic distribution shift. Our method consistently outperforms SOTA methods on both datasets, including:
> > >
> > > - Gen1 Dataset: **+5.5 mAP** over RVT-B (ICCV 2023), **+17.9 mAP** over SNN-ANN (CVPR 2025), **+4.9 mAP** over HsvT-B (ICML 2025), **+3.1 mAP** over EGSST (NeurIPS 2024), **+12.3 mAP** over SpikeYOLO (ECCV 2024)
> > >
> > > - 1Mpx Dataset: **+1.7 mAP** over RVT-B (ICCV 2023), **+22.1 mAP** over SNN-ANN (CVPR 2025), **+0.9 mAP** over EGSST-Y (NeurIPS 2024)
> > >
> > > - **Faster inference** than most transformer-based methods with fewer parameters
> > >
> > > This indicates that our design not only generalizes across sensors and conditions but does so with **better temporal precision and efficiency**.
> > >
> > > ### 3. **Summary of comparison**
> > >
> > > | Method (Venue & Year) | Network Design | Dataset(s) | Innovation Summary |
> > > |------------------------|----------------|------------|---------------------|
> > > | From Chaos Comes Order (ICCV 2023) | Token ordering + CNN | Gen1, 1Mpx | Event token temporal ordering |
> > > | GET (ICCV 2023) | Grouped Transformer | Gen1, N-Caltech101 | Group-wise event token transformer |
> > > | Recurrent ViT (CVPR 2023) | ViT + recurrence | Gen1, 1Mpx | Recurrent vision transformer |
> > > | State Space Models for Event Cameras (CVPR 2024) | Latent state-space model + encoder | Gen1, MVSEC | Sequential latent modeling using SSMs |
> > > | Integer-valued SNN (ECCV 2024) | Integer-only SNN pipeline | Gen1, 1Mpx | Spike-driven object detection with quantized training |
> > > | EGSST (NeurIPS 2024) | GNN + temporal attention | Gen1, 1Mpx | Graph-based spatiotemporal Transformer |
> > > | EventPillars (AAAI 2025) | Sparse voxel pillar encoding + CNN | Gen1, DSEC | Efficient pillar-based encoding for event streams |
> > > | **Ours (NeurIPS 2025)** | CNN + early-stage ConvLSTM + deformable fusion + FPN | Gen1, 1Mpx | Fine-grained temporal modeling, decoupled motion fusion, scale-aware multi-branch encoding |
> > >
> > > As shown, our work introduces a **unique architectural perspective** that combines biological inspiration (early-stage recurrence), motion-aware fusion, and multi-scale event dynamics, achieving **strong performance across diverse sensor settings**.

---

> > > > ### Author Response · Authors · 2025-08-05
> > > >
> > > > (2/2)
> > > >
> > > > ### 4. **Ev-DET200K**
> > > >
> > > > Additionally, we would like to clarify a potential source of confusion regarding the **Ev-DET200K** dataset. Upon closer inspection, we found that each video sequence in this dataset contains **only 5 frames**, resulting in **extremely limited temporal continuity**. This makes it **fundamentally different** from standard event-based detection benchmarks such as **Gen1** and **1Mpx**, where each sequence consists of **continuous streams of events** rather than 5 frames. These standard datasets offer **rich spatiotemporal dynamics** that are essential for evaluating temporal modeling.
> > > >
> > > > Consequently, the **Ev-DET200K** paper reports strong performance for **image-based methods** such as **RetinaNet**, **Mask R-CNN**, and **Swin Transformer**, while **event-specific recurrent methods** like **RVT** perform significantly worse (reported at ~**40 mAP**). In fact, our own **re-implementation of RVT** on this dataset (currently still in training) yields even lower performance (around **20 mAP**), and we are actively investigating the underlying causes.
> > > >
> > > > This observation further supports our view that **Ev-DET200K may not be a suitable benchmark** for evaluating the **temporal modeling capabilities** of event-based detection methods—especially those designed to process **continuous, asynchronous input**. Notably, the **Ev-DET200K** paper also did **not train or evaluate** their method on continuous-time datasets like **Gen1** or **1Mpx**, but only reported **transfer results**, which further aligns with our assessment.
> > > >
> > > > **We appreciate the reviewer’s thoughtful feedback. We hope the above clarification helps demonstrate the novelty and robustness of our approach. We trust the reviewer will make their own informed assessment, and we are open to any further discussion or suggestions.**

---

> > > > > ### Comment · Reviewer_U1DD · 2025-08-06
> > > > >
> > > > > Thanks to the authors for providing the follow-up response to further clarify the issues raised.
> > > > >
> > > > > I have read the clarifications and modifications and find them useful in addressing the concerns. I believe incorporating these aspects into the revised manuscript will largely improve the claims and statements.
> > > > >
> > > > > Therefore, I have changed the rating from borderline reject to borderline accept. However, since I do not directly work on this specific topic, I am still leaning towards maintaining a low confidence score, and leaving more room to the other reviewers in justifying the actual novelty and the completeness of the experiments of this work.
> > > > >
> > > > > Nevertheless, I believe that by including more datasets and having better architectural designs, the framework can be stronger and useful in real-world scenarios.
> > > > >
> > > > > Best,
> > > > >
> > > > > Reviewer U1DD

---

> > > > > > ### Author Response · Authors · 2025-08-06
> > > > > >
> > > > > > Thank you very much for your thoughtful follow-up and for updating your rating. We sincerely appreciate your recognition of our clarifications and your constructive suggestions.
> > > > > >
> > > > > > We will make sure to reflect these aspects and discussions in the revised manuscript, including additional evaluations on datasets (e.g., N-Caltech101), as well as a more detailed analysis of Ev-DET200K to better understand the role of temporal continuity in recurrent temporal modeling. Thank you again for your valuable feedback and encouragement.
> > > > > >
> > > > > > Best regards,
> > > > > > Authors of #1513

---

### Official Review · Reviewer_u4BD · 2025-07-02

**Clarity:** 2
**Significance:** 2
**Originality:** 2
**Rating:** 2
**Confidence:** 5

**Summary:**

This paper addresses the challenge of effectively modeling temporal dynamics for event-based object detection. It proposes a CNN-RNN hybrid architecture that enhances temporal modeling by placing recurrent modules at lower spatial scales, which allows for the capture of fine-grained temporal features in sparse event streams. In addition, the authors introduce a Decoupled Deformable-enhanced Recurrent Layer to decouple per-pixel motion estimation and feature fusion, along with a multi-scale approach for spatio-temporal encoding. The experimental results demonstrate that this method outperforms recent transformer-based models.

**Questions:**

See Weakness

**Ethical Concerns:**

["NO or VERY MINOR ethics concerns only"]

**Final Justification:**

After carefully considering the authors' rebuttal, I still do not find the novelty of the work convincing. Although the authors have introduced some modifications, these appear to be more incremental refinements rather than substantive, novel contributions. Consequently, I maintain my original score.

**Limitations:**

No, it is recommended that the authors explore the limitations of the paper in detail

**Quality:**

2

**Strengths And Weaknesses:**

Strengths:

1. A new hybrid architecture, CNN-RNN, is proposed, offering a new perspective on event-based object detection by prioritizing early-stage temporal modeling.

2. A scale-specific spatio-temporal encoding approach allows for more flexible and comprehensive feature extraction.

Weaknesses:
1. The proposed methods do not present new insights or theoretical advancements in the context of event-based representation learning, particularly in areas like asynchronous data processing or novel feature extraction techniques. This approach primarily improves on existing RNN-based models without introducing a groundbreaking shift in understanding the underlying event-based object detection problem.
2. (1) Lack of recent state-of-the-art event-based object detection methods, such as: SpikeYOLO [1], EGSST [2], EventPillars [3], HsVT [4], SNN-ANN [5], etc；(2) Compared to recent Sstate-of-the-art methods, the performance improvement is limited; (3) Has the proposed method been tested for other alternative event representations (such as time surfaces, histogram)? (4) Can the proposed method be tested on the EvDET200K [6] dataset?

[1] Integer-valued training and spike-driven inference spiking neural network for high-performance and energy-efficient object detection (ECCV2024)

[2] EGSST: Event-based Graph Spatiotemporal Sensitive Transformer for Object Detection (NeurIPS 2024)

[3] EventPillars: Pillar-based Efficient Representations for Event Data (AAAI2025)

[4] Hybrid Spiking Vision Transformer for Object Detection with Event Cameras (ICML2025)

[5] Efficient Event-Based Object Detection: A Hybrid Neural Network with Spatial and Temporal Attention (CVPR2025)

[6] Object detection using event camera: A moe heat conduction based detector and a new benchmark dataset (CVPR2025)

---

> ### Author Rebuttal · Authors · 2025-07-31
>
> **[W1]:** The proposed methods do not present new insights or theoretical advancements in the context of event-based representation learning, particularly in areas like asynchronous data processing or novel feature extraction techniques. This approach primarily improves on existing RNN-based models without introducing a groundbreaking shift in understanding the underlying event-based object detection problem.
>
> **[A1]:** Thank you for your perspective. Our method mainly focuses on spatiotemporal feature extraction approaches from a new direction. Firstly, starting from the design of recurrent layers, we developed a decoupled deformable-enhanced module based on the characteristic that events contain motion information, aiming to reuse the state features in the recurrent module. This module integrates adaptive spatial sampling and temporal modeling capabilities, significantly improving the alignment effect of temporal features. Secondly, considering that low-scale features may contain richer motion details, we introduced recurrent layers at low scales and re-examined the temporal modeling method under scale-specific conditions. As mentioned in the previous work RED, one of the important reasons for placing recurrent layers at high scales is to prevent recurrent layers from dynamically modeling low-level features that are unnecessary for the given task. However, the decoupled deformable-enhanced module we added before the recurrent layers can greatly reduce these unnecessary features through the smoothing effect of deformable convolution (as shown in Supplementary Materials Figures S3 and S4).
>
> - - -
>
> **[W2]:** (1) Lack of recent state-of-the-art event-based object detection methods, such as: SpikeYOLO [1], EGSST [2], EventPillars [3], HsvT [4], SNN-ANN [5], etc; (2) Compared to recent state-of-the-art methods, the performance improvement is limited; (3) Has the proposed method been tested for other alternative event representations (such as time surfaces, histogram)? (4) Can the proposed method be tested on the EvDET200K [6] dataset?
>
> **[A2]:**
>
> | method    | Gen1 mAP(%) | 1Mpx mAP(%) |
> | --------- | ----------- | ----------- |
> | SpikeYOLO | 40.4        | -           |
> | EGSST-E   | 49.6        | 50.2        |
> | EGSST-E-Y | 47.8        | 48.3        |
> | HsvT-B    | 47.8        | -           |
> | SNN-ANN   | 35          | 27          |
> | Ours      | 52.7        | 49.1        |
>
> Thank you for mentioning many recent and meaningful SOTA methods. Compared with these SOTA methods you referred to, our method still remains competitive, especially in comparison with SpikeYOLO, HsvT-B, and SNN-ANN. EGSST-E has shown good performance on both the Gen1 and 1Mpx datasets. We believe this is partly attributed to its proposed dynamic label enhancement strategy, which improves the accuracy and adaptability of annotations by dynamically adjusting the time window for label matching. Additionally, it is worth noting that when the YOLO detection head is used to replace the RT-DETR detection head (i.e., EGSST-E-Y), the performance of EGSST decreases. Regarding (3) and (4), we are currently training on the EvDET200K dataset using histogram representation, and will share the results immediately during the discussion once available.

---

### Official Review · Reviewer_vrrf · 2025-07-03

**Clarity:** 3
**Significance:** 3
**Originality:** 3
**Rating:** 5
**Confidence:** 4

**Summary:**

This paper proposes a hybrid CNN–RNN architecture for event-based object detection that emphasizes fine-grained temporal encoding at early stages. Raw event streams are discretized into 5 temporal bins over 50 ms, then processed by a shared stem and three Decoupled Deformable-enhanced Recurrent Layers (DDRLs). Each DDRL combines a ConvLSTM with a modulated deformable convolution to align per-pixel motion before spatial downsampling. Multi-scale spatiotemporal features from three resolution branches are fused via an FPN and passed to a YOLOv6-based detection head. Experiments on the Gen1 (304×240) and 1 Mpx (1280×720) driving datasets show state-of-the-art mAPs of 52.7% and 49.1%, respectively, with comparable parameter count and FLOPs.

**Questions:**

1. Have you conducted an ablation over voxelization granularity (e.g., bins ∈ {5, 8, 10}, window ∈ {20 ms, 50 ms, 100 ms}) to confirm that the chosen 5-bin, 50 ms discretization sufficiently captures the fastest event dynamics?
2. Could you visualize the learned deformable offsets (∆xₖ), such as heat maps or flow fields, to validate that the deformable module aligns motion as intended?
3. Please compare or discuss your approach relative to SODFormer (Li et al., 2023) and the Time-Surface Pyramid (Manderscheid et al., 2019). Benchmarking against these fine-grained temporal-encoding methods would help clarify the paper’s novelty and positioning.
4. Can you report end-to-end inference latency, peak GPU memory usage (batch 1), and a small/medium/large object AP breakdown on both Gen1 and 1 Mpx inputs, measured on specified hardware?
5. Would moving one or two key supplementary results (e.g., the Table S1/S2 ablations or a representative offset heat map) into the main manuscript improve transparency around your architectural choices?

**Ethical Concerns:**

["NO or VERY MINOR ethics concerns only"]

**Final Justification:**

The rebuttal and additional experiments have fully addressed my earlier concerns, particularly regarding voxelization ablation, method positioning, motion offset visualization, smoothing analysis, and efficiency breakdowns. These additions strengthen the empirical evidence and clarify the paper’s contributions. In light of the clarifications and new results, I am increasing my rating to 5 (Accept).

**Limitations:**

Yes

**Paper Formatting Concerns:**

None.

**Quality:**

3

**Strengths And Weaknesses:**

Strengths:
1. The early-stage ConvLSTM effectively captures temporal cues before significant spatial pooling, preserving fine-grained motion information.
2. Combining ConvLSTM with a modulated deformable convolution in each DDRL adaptively aligns pixel-wise motion, reducing blur in fast-moving regions.
3. The divide-and-conquer architecture cleanly separates motion estimation from feature fusion, which suits the asynchronous nature of event data.
4. The model delivers state-of-the-art mAP on both Gen1 and 1 Mpx driving benchmarks, demonstrating robustness across resolutions.
5. Comprehensive ablation studies (e.g., DDRL placement, branch design) substantiate key architectural choices.

Weaknesses:
1. The fixed 5-bin, 50 ms voxelization may not capture the fastest or smallest event dynamics, yet no ablation evaluates finer temporal discretization.
2. The related work omits key fine-grained temporal-encoding methods (e.g., SODFormer, Time-Surface Pyramid), leaving the paper’s positioning in this space unclear.
3. There is no visualization or quantitative analysis of the learned offsets (∆xₖ), nor any comparison to optical-flow ground truth to validate that motion alignment is actually occurring.
4. Combining deformable convolution with ConvLSTM could over-smooth spatial detail, but no effective-receptive-field or smoothing analysis is provided.
5. Independently processed branches are fused via a standard FPN without any temporal-state synchronization mechanism, and no small/medium/large AP breakdown or branch-ablation confirms the benefits for small or rapid objects.
6. The efficiency evaluation is incomplete: “Time (ms)” in Table 1 is ambiguous, and there are no hardware-specific end-to-end latency, peak memory, or throughput measurements to support the “compact yet accurate” claim.

---

> ### Author Rebuttal · Authors · 2025-07-31
>
> **[W1, Q1]:** The fixed 5-bin, 50 ms voxelization may not capture the fastest or smallest event dynamics, yet no ablation evaluates finer temporal discretization.
>
> **[A1]:** Thank you for your valuable comments. We have added an ablation study on the number of voxel bins and the temporal window size. To reduce data generation and training time, we randomly selected one-third of the Gen1 training set for training and evaluated on the full Gen1 test set. All other settings were kept consistent with those in the main paper. Our experiments are divided into two groups:
>
> - **Fixed number of bins (5), varying temporal window sizes (20ms, 50ms, 100ms):**
>   The results show that a 50ms window yields relatively better performance. This may explain why several prior works such as RED, RVT, and SSM have also adopted 50ms as the default window size.
>
> - **Fixed temporal window size (50ms), varying number of bins (5, 8, 10):**
>   The results indicate that increasing the number of bins does not lead to significant performance gains. This may be due to the sparse nature of event data—when using voxel representation, 5 channels are already sufficient to capture most of the meaningful information available in the event stream.
>
> | bins | window (ms) | mAP  | mAP50 |
> |------|-------------|------|--------|
> | 5    | 50          | 45.4 | 74.3   |
> | 5    | 20          | 44.9 | 74.7   |
> | 5    | 100         | 43.6 | 72.0   |
> | 8    | 50          | 45.0 | 74.0   |
> | 10   | 50          | 45.4 | 75.2   |
>
> ---
>
> **[W2, Q3]:** Please compare or discuss your approach relative to SODFormer (Li et al., 2023) and the Time-Surface Pyramid (Manderscheid et al., 2019). Benchmarking against these fine-grained temporal-encoding methods would help clarify the paper’s novelty and positioning.
>
> **[A2]:** Thank you for your suggestion. Based on your feedback, we have improved the Related Work section. Compared to SODFormer, both methods aim to model temporal dynamics in event data, but differ in design:
> SODFormer performs event - frame fusion using asynchronous attention to compensate for frame degradation under fast motion or low light. Our method focuses on event - only modeling, emphasizing early - stage temporal encoding to capture fine - grained motion without relying on frame input. SODFormer models long - term dependencies in deep layers via global attention, which is computationally heavy. In contrast, we apply recurrent modeling at lower feature stages, better preserving temporal cues like fast edge motions that tend to degrade after downsampling. Regarding Time - Surface Pyramid, it targets corner detection by building velocity - invariant patterns through local decay, aiming to normalize static temporal changes. Our DDRL module, by combining deformable convolution and recurrence, directly models motion dynamics and better serves event - based object detection tasks
>
> ---
>
> **[W3, Q2]:** There is no visualization or quantitative analysis of the learned offsets ($\Delta$x ), nor any comparison to optical - flow ground truth to validate that motion alignment is actually occurring.
>
> **[A3]:** Thank you very much for your suggestion. We fully agree that visualizing the learned offsets of the deformable convolution is important for better understanding its behavior. We have visualized the offsets from the first deformable - enhanced block using heatmaps. We chose this layer because it retains a higher spatial resolution and the features are less abstract, making the offsets more interpretable. The visualization reveals that the offsets primarily capture motion information within the current frame. And we will include these offset visualizations.
>
> ---
>
> **[W4]:** Combining deformable convolution with ConvLSTM could over-smooth spatial detail, but no effective-receptive-field or smoothing analysis is provided.
>
> **[A4]:** You make a valid point — combining deformable convolution with ConvLSTM may indeed risk over-smoothing spatial details. To investigate this, we provided a visual comparison in the supplementary material (Figures S3 and S4) between features output with and without deformable convolution in the recurrent layers. The results show that while adding deformable convolution introduces some smoothing, it primarily suppresses irrelevant or noisy information that does not contribute to object detection. We will further enhance this analysis by adding more visualizations.
>
> ---
>
> **[W5, W6, Q4]:** Can you report end-to-end inference latency, peak GPU memory usage (batch 1), and a small/medium/large object AP breakdown on both Gen1 and 1 Mpx inputs, measured on specified hardware?
>
> **[A5]:** Thank you for your suggestion. We re-conducted the tests on an Nvidia RTX 3090, comparing with advanced methods such as RVT and SSM. The batch size was set to 1, and the results on the Gen1 dataset are as follows:
>
> | method | mAP(%) | AP_S(%) | AP_M(%) | AP_L(%) | time(ms) | peak memory(MB) |
> | ------ | ------ | ------- | ------- | ------- | -------- | --------------- |
> | RVT-B  | 43.1   | 34.1    | 50.4    | 43.5    | 8.07     | 435.14          |
> | RVT-S  | 42.6   | 33.3    | 50.3    | 46.2    | 8.34     | 358.82          |
> | RVT-T  | 40.2   | 30.8    | 47.8    | 44.6    | 7.18     | 309.38          |
> | S5-B   | 45.7   | 36.5    | 53.1    | 51.1    | 9.19     | 1306.02         |
> | S5-S   | 44.9   | 35.3    | 52.8    | 45.4    | 9.36     | 948.69          |
> | ERGO   | 50.6   | 42.0    | 58.0    | 52.8    | 205      | 1351.61         |
> | Ours   | 52.7   | 45.0    | 59.4    | 51.8    | 8.08     | 434.58          |
>
> Here, AP_S, AP_M, and AP_L represent the detection accuracies for small, medium, and large objects, respectively. It can be seen that our method achieves the best performance while ensuring high inference speed and low GPU peak memory, especially in small object detection.
>
> ---
>
> **[Q5]:** Would moving one or two key supplementary results (e.g., the Table S1/S2 ablations or a representative offset heat map) into the main manuscript improve transparency around your architectural choices?
>
> **[A6]:** Thank you very much for your valuable suggestion. We fully agree with your idea and will make some adjustments.

---

> > ### Comment · Reviewer_vrrf · 2025-08-06
> >
> > Thank you for the rebuttal. All my concerns have been addressed.

---

### Official Review · Reviewer_S6Df · 2025-07-06

**Clarity:** 4
**Significance:** 3
**Originality:** 3
**Rating:** 5
**Confidence:** 4

**Summary:**

This paper considers object detection using event-camera data. The authors argue that temporal information needs to be captured at low spatial scales before the extraction of high-dimensional convolutional features reduces the spatial resolution. The key idea is to introduce event-recurrent (RNN) modules at low spatial scales first, followed by scale-specific (CNN) feature extraction and a feature pyramid network (FPN) for object detection. The proposed RNN-CNN architecture operates over a 3D spatio-temporal voxel grid of event polarities, applies decoupled deformable convolution to fuse the current frame features with historical features, a ConvLSTM layer to capture spatio-temporal features from consecutive event frames, and finally an FPN to fuse multi-scale features. The method is evaluated on two of the widely used datasets for event-based object detection: Gen1 and 1Mpx in comparison to several recent Transformer, Transformer+RNN, GNN, and CNN+RNN architectures.

**Questions:**

1. My most critical suggestion is to add a generalization experiment, e.g., train on one dataset and test on the other. For example, this paper https://arxiv.org/abs/2505.12908 does this in their Table 2. It would be nice to see the performance of the proposed method in a similar experiment.

2. The paper is repetitive about the high-level points that temporal patters in event data appear in low-level features and are best captured before spatial abstraction. This point and several related ones are made multiple times in the Intro, Related Works, Method sections. On the other hand, the review of related work is quite brief and very few qualitative results are presented. I recommend that the authors reduce the repetitiveness of the high-level motivation for the approach and instead add more of: technical details, review of additional related works, qualitative results.

3. The authors should include the time and accuracy results for all methods in Table 1. Also, the best and second-best method should be highlighted for the time results as well.

4. Minor comments:
  - The meaning of hidden state, cell state, and BN are not clear the first time they are introducing in eq. (2), (3). These elements are not explained clearly until Sec. 3.4
  - The references have multiple problems that need to be addressed, e.g., [7] is missing a publication venue, [10] includes et al. in the author list (not preferable), [17], [18], etc. include the publication year and IEEE twice (repetition is unnecessary), etc.

**Ethical Concerns:**

["NO or VERY MINOR ethics concerns only"]

**Final Justification:**

This is a well written paper that proposes a simple and effective architecture for processing event camera data using recurrent layers at low spatial scales, followed by CNN feature extraction and fusion of current and historical features from consecutive frames. The proposed architecture performs very well on the Gen1 and 1MPx datasets in terms of accuracy and speed and is supported by thorough ablation experiments. In the rebuttal, the authors offered to add results which show generalization of the deformable convolution in the current layer design from one training set to a different test set. This is an important supplement to the evaluation. Overall, I think this is a strong paper and the authors have responded well and in detail to the reviewers' questions.

**Limitations:**

Yes

**Quality:**

4

**Strengths And Weaknesses:**

## Strengths:
 + The proposed architecture is simple, well motivated, and performs well. I find the proposed design using several recurrent layers with deformable convolution and LSTM for the temporal events simple and effective. The consideration of both a hidden state and a memory cell in the LSTM model is nice.

 + The performance of the method on the Gen1 and 1Mpx datasets is very good in terms of both accuracy and time in comparison to strong recent baselines.

 + The ablation study results are interesting and, importantly in my opinion, underscore that the proposed design is effective, even without deformable convolution or Squeeze-Excite blocks in the model, and in comparison to Transformer-RNN [13].


## Weaknesses:
 - The use of deformable convolution in the recurrent layer design raises questions about generalization. It is important that the authors evaluate the performance of the model when trained on one dataset and tested on a different one.

 - The presentation in Sec. 1-3 is somewhat repetitive regarding the proposed design. Reducing the repetitiveness will allow more space for presenting additional technical details, adding more quantitative (generalization) and qualitative results, and improving the qualitative results.

 - The qualitative results in Fig. 5 and 6 are hard to see and offer limited value. What are we supposed to observe? How does the visualization show anything about the feature processing and integration by the different models? The evaluation can be strengthened by providing better/additional qualitative results.

---

> ### Author Rebuttal · Authors · 2025-07-31
>
> **[W1, Q1 generalization experiment]:** The use of deformable convolution in the recurrent layer design raises questions about generalization. It is important that the authors evaluate the performance of the model when trained on one dataset and tested on a different one.
>
> **[A1]:**
>
> Thank you for raising this important question regarding the generalization of deformable convolution in the recurrent layer design. We agree that cross-dataset evaluation is essential to verify the model's generalization ability. We conducted generalization experiments using our model, carrying out cross-dataset generalization between the Gen1 and 1Mpx datasets, and compared them with two advanced Transformer and Mamba models (RVT, SSM). We directly used the weights trained on one dataset for testing on the other dataset without fine-tuning the weights.
>
> It can be observed that despite using deformable convolution to dynamically learn offsets in event data, the generalization ability of our model is still comparable to, or even better than, the other two types of methods, especially in the generalization experiment from Gen1 to 1Mpx.
>
> **Generalization: 1Mpx → Gen1**
>
> | method | mAP  | mAP50 |
> |--------|------|--------|
> | Ours   | 12.2 | 35.6   |
> | RVT-B  | 12.4 | 34.0   |
> | S5-B   | 11.8 | 31.5   |
>
> **Generalization: Gen1 → 1Mpx**
>
> | method | mAP  | mAP50 |
> |--------|------|--------|
> | Ours   | 28.1 | 57.4   |
> | RVT-B  | 25.9 | 50.6   |
> | S5-B   | 21.6 | 42.9   |
>
> ---
>
> **[W2, Q2]:** The paper is repetitive about the high-level points that temporal patterns in event data appear in low-level features and are best captured before spatial abstraction. This point and several related ones are made multiple times in the Intro, Related Works, Method sections. On the other hand, the review of related work is quite brief and very few qualitative results are presented. I recommend that the authors reduce the repetitiveness of the high-level motivation for the approach and instead add more of: technical details, review of additional related works, qualitative results.
>
> **[A2]:** Thank you very much for your constructive suggestion. Based on your feedback, we will make the following improvements:
>
> - **Reducing redundancy:** We will revise and streamline the Introduction and Related Work sections to eliminate repetitive statements about our method. The discussion on the decoupled deformable-enhanced recurrent layer and low-level spatiotemporal modeling will be condensed and clarified within the Method section.
>
> - **Expanding related work:** The original Related Work section was kept concise due to space limitations. We will expand this section to include recent works such as ERGO-12 and S5-ViT, providing a more comprehensive overview of event data modeling approaches.
>
> - **Qualitative visualization analysis:** Figures 5 and 6 present qualitative results from our ablation study to demonstrate the effectiveness of our proposed module. We acknowledge that the detection boxes may have been too thin, making it difficult to observe differences. In several scenes, other ablated variants exhibit missed detections or false positives.
>
> ---
>
> **[W3, Q3 Table1]:** The authors should include the time and accuracy results for all methods in Table 1. Also, the best and second-best method should be highlighted for the time results as well.
>
> **[A3]:** The missing entries in Table 1 correspond to earlier works published before the release of the 1Mpx dataset, and these methods generally exhibit lower performance. Additionally, their original papers did not report inference time, which makes it difficult for us to provide fair or reproducible runtime comparisons.
>
> ---
>
> **[Q4]:** Minor comments
> **[A4]:** Thank you for your careful reading. We have addressed your suggestions one by one and thoroughly checked the manuscript for any other potential issues.

---

### Note · Authors · 2025-08-13

Dear Area Chair and Reviewers,

Thank you all for your thoughtful reviews and active participation in the discussion. We are truly grateful for your recognition of our work and the valuable suggestions, which we will fully incorporate into the final revision.

We respectfully note that **Reviewer u4BD has not yet responded to our rebuttal**, while **all other reviewers have already updated their final scores**. We have carefully addressed all the concerns raised during the rebuttal phase.

As today marks the final day of the AC-Reviewer discussion period, we would sincerely appreciate it if Reviewer **u4BD** could kindly **take a moment to consider our responses and adjust the score if appropriate**.

Best regards,
Authors of Submission #1513

---

### Decision · Program_Chairs · 2025-09-17

**Decision:**

Accept (poster)

**Comment:**

This paper proposes a hybrid CNN–RNN architecture for event-based object detection for modeling fine-grained temporal features. The main technical contribution is the Decoupled Deformable-Enhanced Recurrent Layer (DDRL), a module that separates per-pixel motion estimation from feature fusion. This is implemented using event-recurrent (RNN) modules at low spatial scales first, followed by scale-specific (CNN) feature extraction and a feature pyramid network (FPN) for object detection.

Most of the reviewers commended the strong empirical results, clear writing and extensive ablations. The reviewers did raise some concerns about the novelty, and comparison with SOTA, and all the reviewers have suggested that these should be included in the final version. The authors are encouraged to use this feedback to further improve the paper.
Given the overall positive feedback from the reviewers, we propose an Accept for this paper.